# Learning the Language of Protein Structure

**Benoit Gaujac**[*,1]**, Jérémie Donà**[*,1]**,**
**Liviu Copoiu**[1]**, Timothy Atkinson**[1]**, Thomas Pierrot**[1] **and Thomas D. Barrett**[1]
[*]Equal contributions: {b.gaujac, j.dona}@instadeep.com, [1]InstaDeep

**Reviewed on OpenReview:** https://openreview.net/forum?id=SRRPQIOS4w

## Abstract

Representation learning and *de novo* generation of proteins are pivotal computational biology tasks. Whilst natural language processing (NLP) techniques have proven highly effective for protein sequence modelling, structure modelling presents a complex challenge, primarily due to its continuous and three-dimensional nature. Motivated by this discrepancy, we introduce an approach using a vector-quantized autoencoder that effectively tokenizes protein structures into discrete representations. This method transforms the continuous, complex space of protein structures into a manageable, discrete format with a codebook ranging from 432 to 64000 tokens, achieving high-fidelity reconstructions with backbone root mean square deviations (RMSD) of approximately 1-4 Å. To demonstrate the efficacy of our learned representations, we show that a simple GPT model trained on our codebooks can generate novel, diverse, and designable protein structures. Our approach not only provides representations of protein structure, but also mitigates the challenges of disparate modal representations and sets a foundation for seamless, multi-modal integration, enhancing the capabilities of computational methods in protein design.

## 1 Introduction

The application of machine learning to large-scale biological data has ushered in a transformative era in computational biology, advancing both representation learning and *de novo* generation. Particularly, the integration of machine learning in molecular biology has led to significant breakthroughs (Sapoval et al., 2022; Chandra et al., 2023; Khakzad et al., 2023; Jänes and Beltrao, 2024), spanning many complex and inhomogenous data modalities, from sequences and structures through to functional descriptors and experimental assays, many of which are deeply interconnected and have attracted significant modelling efforts.

Currently, the deep learning landscape is increasingly converging towards a unified paradigm centered around attention-based architectures (Vaswani et al., 2017) and sequence modeling. This shift has been driven by the impressive performance and scalability of transformers, and even accelerated since treating non-standard data modalities as sequence-modeling problems has proven highly effective. Indeed, transformer-based models are leading methods in many machine learning domains, including image representation (Radford et al., 2021), image (Chen et al., 2020; Chang et al., 2022) and audio generation (Ziv et al., 2024), and for reinforcement learning (Chen et al., 2021; Boige et al., 2023). This trend has greatly benefited sequence-based biological models, allowing for the direct application of NLP methodologies like GPT (Radford and Narasimhan, 2018) and BERT (Devlin et al., 2019) with notable success (Ferruz et al., 2022; Lin et al., 2023b).

In particular, large multi-modal models (LMMs), leveraging transformer backbones, are emerging as a key tool with applications in various fields such as: ubiquitous AI (GPT4 (Achiam et al., 2023), LLaVA (Liu et al., 2024b), Gemini (Gemini et al., 2023), Flamingo (Alayrac et al., 2022)); text-conditioned generation of images (Parti (Yu et al., 2022b), Muse (Chang et al., 2023)) or sounds (MusicGen (Copet et al., 2023)); and even reinforcement learning (Reed et al., 2022). LMMs are also instantiated in biological settings, such as medicine (MedLlama (Xie et al., 2024), Med-Gemini (Yang et al., 2024b) Med-PaLM (Tu et al., 2024)) and genomics (ChatNT (Richard et al., 2024)). Core to all these LMMs is the use of pre-trained encoders as a

mechanism for combining modalities in sequence space. For instance, LLaVA (Liu et al., 2024b) and Flamingo (Alayrac et al., 2022) used pre-trained vision encoders, ViT-L/14 (Radford et al., 2021) and Normalizer Free ResNet (NFNet) (Alayrac et al., 2020) respectively, while Copet et al. (2023) leverages the codec of Défossez et al. (2023). In general, training robust representations of specific modalities facilitates their incorporation into state-of-the-art LMMs. However, to the best of our knowledge, there is no established methodology that readily allows the application of sequence-modelling to protein structures.

Despite these advances in related domains, structure-based modeling of biological data such as proteins remains a formidable challenge. Unlike sequences, protein structures are inherently three-dimensional and continuous, which complicates the direct application of transformer models that primarily handle discrete data. Instead, structure-based methods often design bespoke geometric deep learning methodologies to process Euclidean data; for example, graph-neural network encoders (Dauparas et al., 2022; Krapp et al., 2023) and structurally aware modules such as those in AlphaFold (Jumper et al., 2021). Moreover, generative modelling of structures is typically performed with methods designed for continuous variables; such as diffusion (Watson et al., 2023; Yim et al., 2023) and flow matching (Bose et al., 2024), rather than the discrete-variable models that have proved so successful in sequence modelling.

In this work, we aim to address this gap by learning quantized representations of protein structures enabling to efficiently leverage sequence-based language models. The key objectives of this work are:

(i) **To convert protein structures into the discrete domain** – We propose the transformation of structural information of proteins into discrete sequential data, enabling seamless integration with sequence-based models.

(ii) **To learn a discrete and potentially low-dimensional latent space** – By learning a discrete latent space through finite scalar quantization, we facilitate the mapping of continuous structures to a finite set of vectors. This effectively builds a vocabulary for protein structures, and can be pushed into low dimensions for applications with limited resources.

(iii) **To achieve a low reconstruction error** – We aim to minimize the reconstruction error of the learned discrete representation, typically within the range of 1-4 Ångströms.

Our contributions are threefold. First, we introduce a series of quantized autoencoders that effectively discretize protein structures into sequences of tokens while preserving the necessary information for accurate reconstruction. Second, we validate our autoencoders through qualitative and quantitative analysis, and various ablation studies, supporting our design choices. Third, we demonstrate the efficacy and practicality of the learned representations with experimental results from a simple GPT model trained on our learned codebook, which successfully generates novel, diverse, and structurally viable protein structures. We release all experimental code at `https://github.com/instadeepai/protein-structure-tokenizer/` and the trained model weights at `https://huggingface.co/InstaDeepAI/protein-structure-tokenizer/`.

## 2 Method

### 2.1 Protein Structure Autoencoder

Our objective is to train an autoencoder that maps protein structures to and from a discrete latent space of sequential codes. Following prior works (Yim et al., 2023; Wu et al., 2024), we consider the backbone atoms of a protein, $N - C_\alpha - C - O$, to define the overall structure.

For a protein consisting of $N$ residues, we seek to map its structure, represented by the tensor of the backbone atoms coordinates $\mathbf{p} \in \mathbb{R}^{N \times 4 \times 3}$, to a latent representation $\tilde{\mathbf{z}} = [\tilde{\mathbf{z}}_1, \ldots \tilde{\mathbf{z}}_{\frac{N}{r}}]$, where a $r$ is a downsampling ratio, controlling the size of the representation. Note that each element $\tilde{\mathbf{z}}_i$ can only take a finite number of values, with the collection of all possible values defining a codebook $\mathcal{C}$.

A schematic overview of the our autoencoder is depicted in Figure 1. In this section, we focus on the three components of the model; the *encoder* $e_\theta$ extracting a set of $\frac{N}{r}$ embeddings of dimension $c$ denoted $\mathbf{z} \in \mathbb{R}^{\frac{N}{r} \times c}$

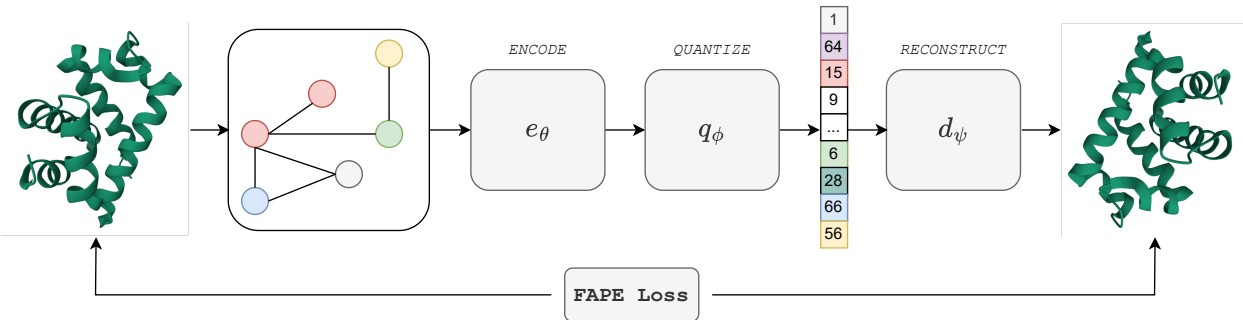

Figure 1: Schematic overview of our approach. The protein structure is first encoded as a graph to extract features from using a GNN. This embedding is then quantized before being fed to the decoder to estimate the positions of all backbone atoms.

, the *quantizer* $q_\phi$ that discretizes $\mathbf{z}$ to obtain a quantized representation $\tilde{\mathbf{z}}$, and the *decoder* $d_\psi$ that predicts a structure $\tilde{\mathbf{p}} \in \mathbb{R}^{N \times 4 \times 3}$ from $\tilde{\mathbf{z}}$. The learnable parameters are respectively denoted $(\theta, \phi, \psi)$ and the learning setting summarizes as:

$$\mathbf{p} \xmapsto{e_\theta} \mathbf{z} \xmapsto{q_\phi} \tilde{\mathbf{z}} \xmapsto{d_\psi} \tilde{\mathbf{p}}.$$

**Encoder** The encoder maps the backbone atoms positions $\mathbf{p} \in \mathbb{R}^{N \times 4 \times 3}$, to a continuous downsampled continuous representation $\mathbf{z} \in \mathbb{R}^{\frac{N}{r} \times c}$ where $r$ is the downsampling ratio:

$$e_\theta : \mathbf{p} \in \mathbb{R}^{N \times 4 \times 3} \mapsto \mathbf{z} \in \mathbb{R}^{\frac{N}{r} \times c} \tag{1}$$

Note that when the downsampling ratio $r$ is set to 1, each component $\mathbf{z_i} \in \mathbb{R}^c$ can be interpreted as the encoding of residue $i$.

This representation learning task is similar to the traditional task of mapping point-clouds to sequences (Yang et al., 2024a; Boget et al., 2024). Inverse folding (Ingraham et al., 2019; Dauparas et al., 2022) is another example, that aims at estimating the sequence of amino acids corresponding to a given a protein structure. Recently, ProteinMPNN has shown remarkable capacity at the inverse folding task. We follow their design choices and parameterize our encoder using a Message-Passing Neural Network (MPNN) (Dauparas et al., 2022). In addition, we introduce a cross-attention mechanism, detailed in Appendix A.1 and Algorithm 2, enabling us to effectively compress the representation of the structure by a downsampling ratio $r$. We maintain locality using a custom attention masking scheme in the downsampling layer (see Appendix A.1), ensuring that each downsampled node aggregates information from a small number of neighboring nodes in the original sequence space.

MPNN operates on a graph consisting of a set of vertices and edges. We follow Dauparas et al. (2022); Ganea et al. (2022) and set as initial node features a positional encoding that reflects the residue's ordering within the sequence, while for the edge features, we use a concatenation of pairwise distance features, relative orientation features and relative positional embeddings. Note that as positional encoding, relative distance, and orientation are invariant with respect to the frame of reference, the input data fed to the model are invariant to rotation and translation. This invariant encoding of the input structures guarantees the invariance of the learned representation regardless of the chosen downstream architecture. The encoding scheme is described in detail in Appendix A.2.

**Quantization** The quantizer plays a crucial role in our work by discretizing all continuous latent representations into a sequence of discrete codes. Traditional methods typically involve direct learning of the codebook (van den Oord et al., 2017; Razavi et al., 2019). However, in line with the literature, we suffered several drawbacks associated with these approaches. Indeed, explicit vector quantization is particularly expensive as it involves computation of pairwise distances, particularly problematic for long sequences and large codebooks. Moreover, we faced training instabilities that arise from both the bias in the straight-through estimator and the under-utilization of the codebook capacity, often referred to as codebook collapse, make learning a

discretized latent representation a hard optimization problem (Huh et al., 2023; Takida et al., 2024). To address these challenges, we leverage the recent Finite Scalar Quantization (FSQ) framework (Mentzer et al., 2024), which effectively resolves the aforementioned issues, notably by reducing straight-through gradient estimation error.

FSQ learns a discrete latent space by rounding a bounded low-dimensional encoding of the latent representation. Consider $\mathbf{z_i} \in \mathbb{R}^c$, the $i^{th}$ element of the continuous encodings $\mathbf{z}$, and the quantization levels $\mathbf{L} = [L_1, \ldots, L_d] \in \mathbb{N}^d$. Its discretized counterpart, denoted as $\tilde{\mathbf{z}}_\mathbf{i} \in \mathbb{Z}^d$, typically with $d \le 8$, is defined by:

$$\tilde{\mathbf{z}}_\mathbf{i} = \text{round}\left(\frac{\mathbf{L}}{2} \odot \tanh(W\mathbf{z_i})\right) \tag{2}$$

where $\odot$ the element-wise multiplication and $W \in \mathbb{R}^{d \times c}$ is a projection weight matrix (see Appendix A.1 for more details). In doing so, each quantized vector $\tilde{\mathbf{z}}_\mathbf{i}$ can be mapped to an index in $\{1, \ldots, \prod_{j=1}^d L_j\}$. This implicitly defines a codebook by its indices, where each index is associated a unique combination of the each dimension values. In our implementation, the quantized representation $\tilde{\mathbf{z}} = [\tilde{\mathbf{z}}_\mathbf{0}, \ldots, \tilde{\mathbf{z}}_{\frac{\mathbf{N}}{\mathbf{r}}}]$ is then projected back to the latent space with dimension $c$ (line 5 of Algorithm 1). Optimization is then conducted using a straight-through gradient estimator. Equation (2) ensures that the approximation error introduced by the straight-through estimator is bounded.

**Decoder**  The decoder module of our framework is tasked with estimating a structure $\hat{\mathbf{p}}$ from the latent quantized representation $\tilde{\mathbf{z}}$:

$$d_\psi : \tilde{\mathbf{z}} \in \mathbb{R}^{\frac{N}{r} \times c} \mapsto \tilde{\mathbf{p}} \in \mathbb{R}^{N \times 4 \times 3} \tag{3}$$

The task our decoder addresses formulates more broadly as a sequence to point-cloud task. A paradigmatic example of such a task in biology is the protein folding problem, where the conformation of a protein is estimated given its primary sequence of amino-acids. Jumper et al. (2021) successfully tackled this task proposing a novel architecture for point cloud estimation from latent sequence of embeddings. Therefore, we use AlphaFold-2 structure module to parameterize our decoder and learn its parameters from scratch.

Specifically, the structure module of AlphaFold-2 parameterizes a point cloud using *frames*. A frame is defined by a tuple $\mathbf{T} = (\mathbf{R}, \mathbf{t})$ where $\mathbf{R}$ is the frames' orientation (*i.e.* a rotation matrix) and $\mathbf{t}$ is the frame center (*i.e.* a vector defining the translation of the center of the frame). The origin of the frame is set to the $\mathsf{C}_\alpha$ carbon, and the orientation is defined using the nitrogen and the other carbon atom. For a thorough and mathematical description, we defer to Jumper et al. (2021). However, note that AlphaFold-2's structure module expects both a per-residue representation $(\mathbf{s_i})_{i \le N}$ and a pairwise representation $(\mathbf{k_{i,j}})_{i,j \le N}$ between residues $i$ and $j$. The per-residue representation $\mathbf{s_i}$ is constructed by mirroring the cross-attention mechanism described in Algorithm 2 used for downsampling in order to produce embeddings at the residue level by only varying the size of the initial input queries. The process for constructing the pairwise representation $(\mathbf{k_{i,j}})_{i,j \le N}$ used for reconstruction is described in Algorithm 3 and Appendix A.1.

## 2.2 Training

**Objective**  The Frame Align Point Error (FAPE) loss, introduced in Jumper et al. (2021), is a function that enables the comparison between point clouds. Since there is no guarantee that the coordinates provided by the decoding module are expressed in the same basis as the input structure, direct comparison of the coordinates between the two point clouds becomes challenging. The core concept behind the FAPE loss is to ensure that coordinates are expressed in the same global frame, thereby enabling the computation of the mean squared error. To do so, given a ground-truth frame $\mathbf{T_i}$ and ground-truth atom position $\mathbf{x_j}$ expressed in $\mathbf{T_i}$, and their respective predictions $\mathbf{T_i^P}$ and $\mathbf{x_j^P}$, the FAPE Loss is defined as:

$$\mathcal{L}_{\text{FAPE}} = \|\mathbf{T_i^{P}}^{-1}(\mathbf{x_j^P}) - \mathbf{T_i}^{-1}(\mathbf{x_j})\| \tag{4}$$

In Equation (4), the predicted coordinates of atom $j$ ($\mathbf{x_j^P}$), expressed in the predicted local frame of residue $i$ ($\mathbf{T_i^P}$), are compared to the corresponding true atom positions relative to the true local frame, enabling the joint optimization of both the frames and the coordinates. We found that clamping the FAPE loss with a threshold of 10 improves the training stability.

**Dataset**   We use approximately 310000 entries available in the Protein Data Bank (PDB) (Berman et al., 2000) as training data. The presence of large groups of proteins causes imbalances in the data set, with many proteins from the same family sharing structural similarities. To mitigate this issue, we sample the data inversely proportional to the size of the cluster it belongs to when clustering the data by sequence similarity using MMseqs2[1]. We filter all chains shorter than 50 residues and crop the structures that have more than 512 residues by randomly selecting 512 consecutive amino acids in the corresponding sequences. We randomly select 90% of the clusters for training and use the remaining as test set. Amongst these 10% withheld protein-structure clusters, we retain 20% for validation, the remaining 80% being used for test.

**Model Hyperparameters**   For the encoder, we use a 3 layers message passing neural network following the architecture and implementation proposed in Dauparas et al. (2022) and utilize the `swish` activation function. The graph sparsity is set to 50 neighbors per residue. When the downsampling ratio is $r > 1$, the resampling operation consists of a stack of 3 resampling layers as described in Algorithm 2, the initial queries being defined as positional encodings. We strictly follow the implementation of AlphaFold-2 (Jumper et al., 2021) regarding the structure module and use 6 structure layers.

**Optimization and Training Details**   The optimization is carried out using AdamW (Loshchilov and Hutter, 2019) with $\beta_1 = 0.9$, $\beta_2 = 0.95$ and a weight decay of 0.1. We use a learning rate warm-up scheduler, progressively increasing the learning rate from $10^{-6}$ to $10^{-3}$ over the first 1000 steps, and train the model for 100 epochs on 8 TPU v4-8 with a batch size of 128. With such hyperparameters, the autoencoder model has $4.5M$ parameters, and the training lasts $\sim 32$ hours on a TPU v4-8, which amounts to a total number of FLOP comprised between $7 \times 10^{19}$ and $10^{20}$.

## 3   Experiments

The primary focus of our work is to develop an effective method to encode and quantize 3D structures with high fidelity. In this section we first evaluate, both qualitatively and quantitatively, our vector-quantized autoencoder by considering the compression and reconstruction performance and demonstrate that our tokenizer indeed permits high-accuracy reconstruction of protein sequences. We then further highlight how this can be adapted to downstream tasks, by effectively training a *de novo* generative model for protein structures using a vanilla decoder-only transformer model trained on the next token prediction task.

### 3.1   Autoencoder Evaluation

**Experiments**   We trained six versions of the quantized autoencoder; with small ($K = 4096$ codes) and large ($K = 64000$) codebooks and increasing downsampling ratio ($r$) (from 1 to 4), and in doing so, varying the information bottleneck of our model. For each codebook we evaluate the reconstruction performance achieved on the held out test set, to understand the trade-offs between compression and expressivity associated with these hyperparameter choices. To further assess the compression capacity of our model, we also train two additional quantized autoencoders with smaller codebook sizes, $K = 432$ and $K = 1728$.

**Metrics**   To assess the reconstruction performances of the models, we rely on standard metrics commonly used in structural biology when comparing the similarity of two protein structures. The *root mean square distance (RMSD)* between two structures is computed by calculating the square root of the average of the squared distances between corresponding atoms of the structures after the optimal superposition has been found. The *TM-score* (Zhang and Skolnick, 2005) is a normalised measure of how similar two structures are, with a score of 1 denoting the structures are identical. For context, two structures are considered to have similar fold when their TM-score exceeds 0.5 (Xu and Zhang, 2010) and a RMSD below 2Å is usually seen as approaching experimental resolution.

**Results**   Our results are summarised in Table 1. We find that with a codebook of $K = 64000$ and a downsampling of $r = 1$; our average reconstruction has 1.22 Å RMSD and a TM-score of 0.96. For comparison,

---

[1]The cluster size is readily available in the PDB data set: `https://www.rcsb.org/docs/grouping-structures/sequence-based-clustering`

we also report the reconstruction performance of the exact same models without latent quantization. Whilst increasing the model capacity may allow these scores to be improved even further, this is already approaching the limit of experimental resolution (which is to say, the reconstruction errors are on par with the experimental errors in resolving the structures).

Table 1: Average Test set reconstruction results of our discrete auto-encoding method for several downsampling ratios and (implicit) codebook sizes. For CASP-15 we report the median of the metrics due to the limited dataset size. Note that a RMSD below 2Å is considered of the order of experimental resolution and two proteins with a TM-score $> 0.5$ are considered to have the same fold. The compression is defined as: the number of bits necessary to store the $N \times 4 \times 3$ backbone positions divided by the number of bits necessary to store the $\frac{N}{r}$ tokens multiplied by $\log_2(\# Codes)$

| Downsampling Ratio | Number of Codes | Compression Factor | Results | | CASP15 | |
|---|---|---|---|---|---|---|
| | | | RMSD ($\downarrow$) | TM-Score ($\uparrow$) | RMSD ($\downarrow$) | TM-Score ($\uparrow$) |
| 1 | 432 | 88 | 2.09 Å | 0.91 | 1.75 Å | 0.89 |
| | 1728 | 71 | 1.79 Å | 0.93 | 1.33 Å | 0.94 |
| | 4096 | 64 | 1.55 Å | 0.94 | 1.25 Å | 0.94 |
| | 64000 | 48 | 1.22 Å | 0.96 | 0.94 Å | 0.97 |
| | without quantization | – | 0.97 Å | 0.98 | 1.07 Å | 0.93 |
| 2 | 4096 | 128 | 2.22 Å | 0.90 | 1.73 Å | 0.89 |
| | 64000 | 96 | 1.95 Å | 0.92 | 1.82 Å | 0.90 |
| | without quantization | – | 1.45 Å | 0.95 | 1.44 Å | 0.90 |
| 4 | 4096 | 256 | 4.10 Å | 0.81 | 2.79 Å | 0.77 |
| | 64000 | 192 | 2.96 Å | 0.86 | 2.55 Å | 0.80 |
| | without quantization | – | 2.19 Å | 0.91 | 1.98 Å | 0.84 |

Table 1 clearly indicates that increasing the downsampling factor or decreasing the codebook size correspondingly impacts the reconstruction accuracy. This is expected as it essentially enforces greater compression in our autoencoder leading to a loss of information - nevertheless in all cases we find that the achievable reconstruction performance is still within a few angstrom with TM-scores clearly exceeding the 0.5 threshold on average. The fact that the performances improve with increasing the codebook size also shows that our method does not suffer from codebook collapse (*i.e.* only a subset of the codes being used by the trained model), an issue well known and documented with other quantization methods (Huh et al., 2023; Takida et al., 2024). This is also noticeable in Figure 2 that shows that, given a downsampling ratio, larger codebooks effectively decrease the reconstruction errors. Finally, comparing to continuous autoencoders, our learned quantization demonstrates significant information compression at the expense of only a small decrease in the reconstruction precision of the order of $0.5 - 1$ Å. Additionally, we provide in Appendix A.3 detailed distribution of the RMSD and TM-Scores for both CASP-15 and the held-out test set. Notably, we can see that increasing the downsampling ratio, or decreasing the codebook size tends to thickens the right tail (resp. the left) of the distribution for the RMSD (resp. the TM score).

The reconstruction performance is illustrated in Figure 3, where examples of the model's outputs are superimposed with their corresponding targets in the case of a downsampling factor of $r = 2$ and $K = 64000$ codes. Visually, our model demonstrates its ability to capture the global structure of each protein. Additionally, our model faithfully reconstructs local conformation of each protein preserving essential secondary structures elements of the protein such as the $\alpha$-helices and $\beta$-sheets.

### 3.2 *De novo* protein structure generation

**Experiment** We now demonstrate that our learned discrete autoencoder can be effectively leveraged for downstream tasks. In particular, we consider generation of protein structures from a model trained in our latent space as a paradigmatic demonstration of our tokenizers utility. This is not just because generative models for *de novo* protein design are of great interest for drug discovery – enabling rapid *in silico* exploration

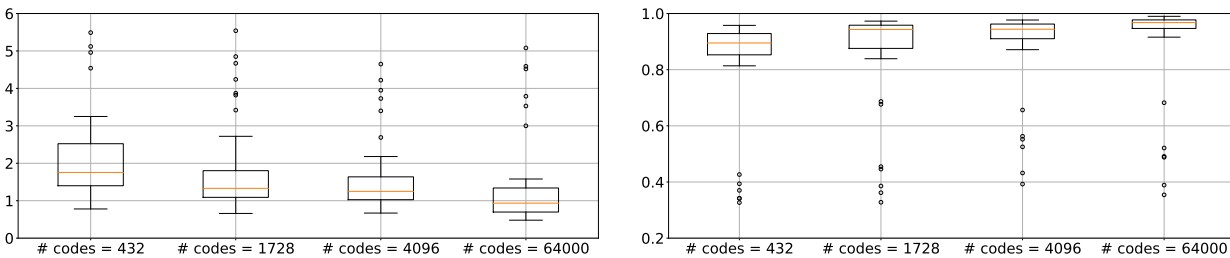

Figure 2: Evolution of the RMSD (left) and TM-score (right) distribution with the codebook size for a downsampling ratio of 1 on CASP-15 data.

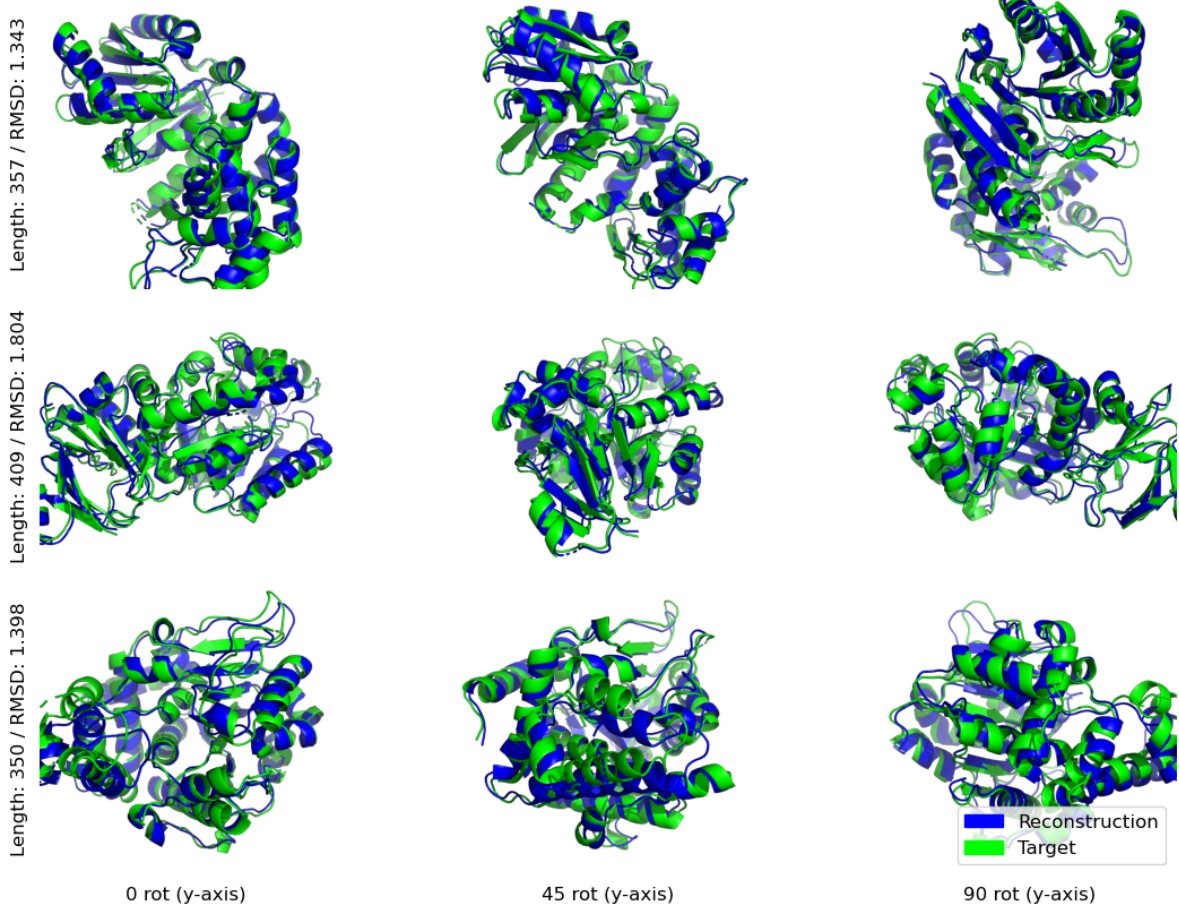

Figure 3: Visualisation of the model reconstruction (blue) super-imposed with the original structures (green) for a downsampling factor of $r = 2$ and $K = 64000$ codes (see Table 1 for detailed results). Each row shows a different structures seen from a different rotation angle (column). The length and reconstruction RMSD are also given on the left of the most left column.

of the design space – but also as it directly leverages our compressed representation of protein structures using established sequence modelling architectures.

Specifically, we tokenize our dataset (defined in Section 2.2; full details on dataset preparation for this experiment can be found in Appendix A.4.1) using a downsampling factor of 1 and a codebook with 4096 codes (first line of Table 1). This choice is motivated by the trade-off between reconstruction performance (this

Table 2: Structure generation metrics for our method alongside baselines (and nature) specifically designed for protein structure generation. Self-consistent TM-score (scTM) and self-consistent RMSD (scRMSD) are two different ways to assess the designability of the generated structure. Note that while high novelty score is desirable, structures that are too far from the reference dataset can also be a sign of unfeasible proteins.

| Method | scTM ($\uparrow$) | scRMSD ($\uparrow$) | Novelty | Diversity |
|---|---|---|---|---|
| Ours | 87.22 % | 61.83% | 23.3% | 60.11 % |
| FrameDiff | 75.77% | 25.31% | 56.64% | 82.0% |
| RFDiffusion | 97.07% | 71.14% | 86.11% | 95.0% |
| Validation Set | 97.67% | 82.36 % | – | 70.1% |

model achieves results close to experimental resolution), extensive datasets (GPT training benefits from large datasets, so we favor a low downsampling ratio, choosing $r = 1$), and parameter efficiency (smaller codebooks imply fewer parameters , we set $K = 4096$). More specifically, we train an out-of-the-box decoder-only transformer model with 20 layers, 16-heads and an embedding dimension of 1024 (344M parameters) on a next-token-prediction task on the training split. This decoder-only model is used to generate new sequences of tokens which are then mapped back to 3D protein structures using our pre-trained decoder.

**Metrics** We evaluate the generated structures on different aspects: *designability*, *novelty* and *diversity*. *Designability*, or *self-consistency*, uses ProteinMPNN (Dauparas et al., 2022) to predict a potential sequence for the generated structure and refolds it using ESMFold (Lin et al., 2022) to report the structural similarity between the original and redesigned structure. We follow the literature and report the proportion of designed proteins with self-consistent TM-score (scTM) above 0.5 respectively the number of designed proteins with self-consistent RMSD (scRMSD) below 2Å. The rationale is that generated structures should be sufficiently natural that established methodologies recognise them and agree on the fundamental biophysical properties.

Moreover, a useful generative model will provide samples that are varied and not simple replications of previously existing structures. To assess this, we measure the *diversity* and *novelty* of our generation. Diversity characterizes the structural similarities between the generated structures and is defined using structural clustering. Specifically, we report the number of clusters obtained when using the TM-score as the similarity metric, normalised by the number of generated sequences. Novelty compares the structural similarity of the generated structure with a dataset of reference. Here again, we use the TM-score as our similarity metric and consider a structure as novel if its maximum TM-score against the reference dataset is below 0.5. As is commonly done, we only report the proportion of novel samples in our generated dataset. Overall, we follow Yim et al. (2023) for the implementation of these metrics and refer the reader to Appendix A.4.2 for more details on the validation pipeline.

**Baselines** To gauge how our structure generation GPT model fares against specifically designed protein design methods, we compare our model with FrameDiff (Yim et al., 2023) and RFDiffusion (Watson et al., 2023). Both use bespoke SE(3) diffusion models specifically designed and trained the structure generation task. Note that, the state-of-the-art RFDiffusion leverages the extensive pre-training of RoseTTAFold (Baek et al., 2021) on a large dataset and requires considerable computational cost. On the contrary, our generation model is a standard GPT and the objective here is to show how one can readily use the tokenized representation learned with our method for protein design.

**Results** The results for the different metrics are given in Table 2 with the sampling strategy used for each method described in Appendix A.4.3. It is noteworthy that while not specifically designed for the protein structure generation task, even our simple GPT model is able to generate protein structures of quality on par with a method specific to protein design such as FrameDiff. For comparison, and to set an upper bound of what to expect in terms of the designability scores, we compute self-consistency and diversity metrics for 1600 randomly sampled structures from our validation and report them in Table 2. While our method shows competitive performance at generative designable domains, the results are more nuanced for the novelty

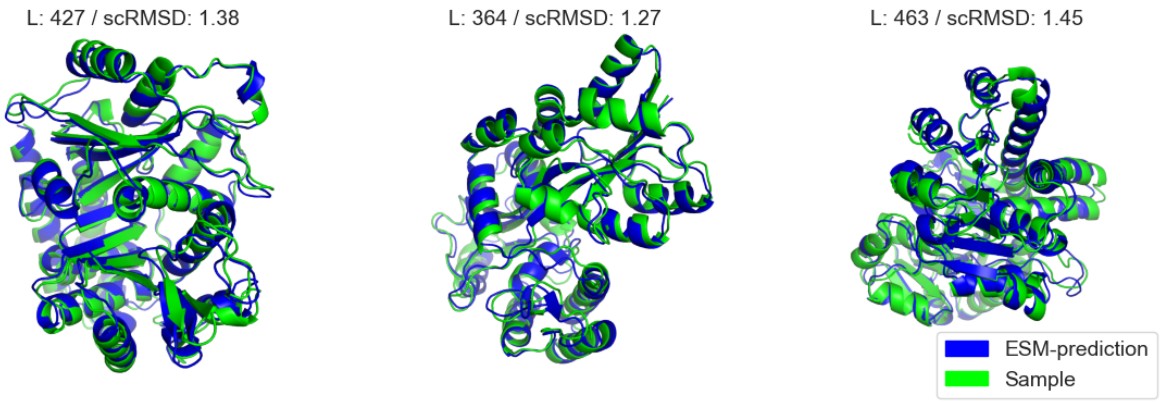

Figure 4: Visualisation of generated samples (green) super-imposed with their self-consistent ESM-predicted structures (blue).

and diversity scores. Especially, our model seems to generate domains that are structurally closer to the reference data set than the ones generated by the baselines. One explanation for this can be found in the sampling method where the chosen parameters favored samples closer to the modes of the data distribution as discussed in Appendix A.4.3. Although generating structures that are novel and diverse is desirable, structures that differ too much from the natural proteins found in the reference dataset, can also indicate unrealistic structures, making the novelty and diversity metrics harder to interpret on their own. Indeed, when visualizing novelty against designability (see Figure 20), we can see that while FrameDiff generates more novel but less designable structures (lower-right corner), our structures are more designable at the expense of lower novelty. In the light of Figure 20, we compute the number of structures with cathTM $< 0.5$ (novel) and scRMSD $< 2\text{Å}$(designable) and find that $9.01\,\%$ (70 out of 777) of our samples are designable and novel relative to $8.18\,\%$ (53 out of 648) for FrameDiff and $58.58\,\%$ (379 out of 647) for RFDiffusion. We provide additional results and analyses in Appendix A.4.4.

In Figure 4, we visualize random samples from our model superimposed with the predicted ESM structure used in the designability metric. We first notice that the generated samples exhibit diverse structure, with non trivial secondary elements ($\alpha$-helices and $\beta$-sheets). We also note that the ESM-predicted structures are closely aligned with the original samples, showing the designability aspect of the generated structures.

## 4 Related Works

**Learning from Protein Structures.** Learning from protein structure is thriving research field that encompasses a wide variety of tasks crucial task. For instance, inverse folding (Ingraham et al., 2019; Hsu et al., 2022; Dauparas et al., 2022; Jing et al., 2021) or binding site estimation (Krapp et al., 2023; Ganea et al., 2022) are critically enabling for drug design (Scott et al., 2016). Others (Consortium, 2006; Robinson et al., 2023; Kucera et al., 2023; Gligorijević et al., 2021) focus on learning from protein structure to provide a better understanding of the role of the structure, hence pushing the knowledge frontier of biological processes.

**Representation Learning of Protein Structures.** Eguchi et al. (2022) learns a VAE using as inputs matrix distances and predicting 3D coordinates. The supervision is done by comparing distance matrices. Despite the straightforward nature of sampling from a VAE latent space, it often yields subpar sample quality, see Kingma et al. (2016).

Discrete representation learning for protein structures has recently garnered increasing attention. Foldseek (van Kempen et al., 2024) introduces a quantized autoencoder to encode local protein geometry, demonstrating success in database search tasks. However, as it focuses solely on local features at the residue level, it lacks the capacity to provide global representation of protein structures. This limitation restricts its application in tasks like structure generation or binding prediction, where global information is critical (Krapp et al.,

2023). Building on the 3Di-alphabet introduced by Foldseek, Su et al. (2024); Heinzinger et al. (2023) propose structure-aware protein language models that integrate structure tokens with sequence tokens. Additionally, Li et al. (2024) combines a structural autoencoder with K-means clustering applied to the latent representation of a fixed reference dataset.

More closely related to our work, Gao et al. (2023) adapts VQ-VAE (van den Oord et al., 2017) for protein structures. However, its limited reconstruction performance constrains its applicability. Lin et al. (2023a) explores discrete structural representation learned by such VQ-VAEs (van den Oord et al., 2017), while Liu et al. (2023) trains a diffusion model on the discrete latent space derived from this approach. Similarly, and concurrently with our work, Liu et al. (2024a) combines finite scalar quantization (Mentzer et al., 2024) with a specialized transformer-based autoencoder for proteins, RNA, and small molecules.

Very recently, efforts have emerged to combine quantized structural representation with discrete sequence representation, enabling multimodal generative models trained on a joint discrete latent space. FoldToken (Gao et al., 2024a;b), is a concurrent approach that shares conceptual similarities with our work but differs in key methodological and application-oriented aspects. FoldToken employs joint quantization of sequence and structure, enabling integration across modalities, whereas our method focuses exclusively on structural information. This decoupling allows for mode-specific pretraining, aligning with strategies from subsequent works (Hayes et al., 2024; Lu et al., 2024). Methodologically, FoldToken introduces a series of improvements to existing VQ methods (van den Oord et al., 2017) aimed at enhancing reconstruction accuracy; whereas we adopt the FSQ framework, which reduces the straight-through gradient estimation gap inherent to VQ methods while improving codebook utilization. Furthermore, while FoldToken primarily emphasizes backbone inpainting and antibody design (Gao et al., 2024b;a), our work considers de novo generation of complete structures.

**Generation of Protein Structures.** A substantial body of literature addresses the challenge of sampling the protein structure space. Numerous studies have advocated for the use of diffusion-based models (Wu et al., 2024; Yim et al., 2023; Watson et al., 2023) or flow matching techniques (Bose et al., 2024). While many of these works, such as those by Yim et al. (2023); Watson et al. (2023); Bose et al. (2024), employ complex architectures to ensure invariance to rigid-body transformations, Wu et al. (2024) opted for an alternative parameterization directly preserving symmetries allowing the authors to capitalize on conventional architectures, yet working only on small protein crops. Recently, Wang et al. (2024b) employs a lookup-free quantizer (Yu et al., 2024) as a structure tokenizer and trains a diffusion protein language model (Wang et al., 2024a) on the concatenated sequence and structure tokens. Other works (Hayes et al., 2024; Lu et al., 2024) simply uses VQ-VAEs (van den Oord et al., 2017) to tokenize the structures and train large language models (LLM) on the combination of sequence and structure tokens.

## 5  Conclusion

This work demonstrates a methodology for learning a discrete representation of a protein geometry; allowing the mapping of structures into sequences of integers whilst still recovering near native conformations upon decoding. This sequential representation not only simplifies the data format but also significantly compresses it compared to traditional 3D coordinates. Our belief is that the primary contribution of our work lies in setting the stage for applying standard sequence-modelling techniques to protein structures.

A prerequisite for such development is the expressiveness of the tokenized representation, which must capture the necessary information to enable high-fidelity reconstruction of the 3D structures. Both empirical and visual inspection confirm this to be the case for our proposed methodology. Indeed, as codebook collapse is effectively mitigated by the use of FSQ, larger codebook vocabularies and an increased tokens usage provide straightforward recipes for improving the reconstruction accuracy by reducing the information bottleneck of the autoencoder.

The first step towards sequence-based modelling of structures is the proof-of-concept GPT model, trained on tokenized PDB entries, that serves as a simple *de novo* generating protein backbones. That the achieved results are competitive with some recent diffusion-based model underlines the promise of this paradigm. While a simple GPT does not yet match seminal approaches like RFDiffusion, it is important to recognize

that the performances of the latter stem from extensive developments in 3D generative modeling. Given the remarkable performance of sequence-modelling algorithms across a diversity of data modalities, equivalent efforts could also provide simpler and more powerful treatments of protein structure. This represents a promising direction for future research based on this work.

## 6  Broader Impact

The discrete protein-structure tokenizer and accompanying generative models presented in this work can advance computational protein engineering in several ways. By converting atomic coordinates into a unified token vocabulary ($432 - 64\,000$ tokens), the framework allows researchers to apply the full range of transformer-based language-model techniques—pre-training, fine-tuning, and prompt-driven generation—directly to structural data. Because the models and weights are openly released in reproducible configurations, laboratories with limited computational resources can replicate our baselines, while larger centres can exploit the highest-fidelity variants. A shared structural vocabulary also facilitates interoperability: it can be incorporated into existing sequence predictors, docking pipelines, or function classifiers without redesigning model inputs, thereby lowering experimental turnaround times for enzyme, antibody, and other protein-based therapeutic design. Potential risks must also be acknowledged. Any generative protein model could, in principle, be misapplied to design toxic or immuno-evasive molecules. In silico structures may be over-interpreted if they are not validated experimentally, leading to misplaced confidence in downstream applications.

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

# A  Appendix

## A.1  Architectures

We provide in this section additional details on the autoencoder architecture.

**Quantizer**  For the FSQ quantizer, we use linear projections for encoding and decoding of the codes following the original work of (Mentzer et al., 2024). For all the experiments, we fix the dimension of the codes to $d = 6$. Then, we quantize each channel into $L$ unique values $L_1, \ldots, L_d$ and refer to the quantization levels as $\mathbf{L} = [L_1, \ldots, L_d]$. The size of the codebook $\mathcal{C}$ is given by the product of the quantization levels: $|\mathcal{C}| = \prod_{j=1}^{d} L_j$. For the experiments with small codebooks with $|\mathcal{C}| = 4096$, we use $\mathbf{L} = [4, 4, 4, 4, 4, 4]$ and for large codebooks experiments, $|\mathcal{C}| = 64000$, we take $\mathbf{L} = [8, 8, 8, 5, 5, 5]$.

In more details the FSQ quantizer writes as Algorithm 1:

---
**Algorithm 1** Finite Scalar Quantization
---
1: **Input:** $\mathbf{z_i} \in \mathbb{R}^c$ (residue embedding), $W_{proj} \in \mathbb{R}^{d \times c}$ $W_{up-proj} \in \mathbb{R}^{c \times d}$ (weight matrices), $\mathbf{L}$ number of levels
2: **Output:** $\tilde{\mathbf{z}}_\mathbf{i}$ (Quantized output)
   // Compute low dimensional embedding
3: $\mathbf{z_i} = W_{proj} \cdot \mathbf{z_i}$
   // Bound each element $z_{ij}$ between $[-L_j/2, L_j/2]$
4: $z_{ij} = \frac{L_j}{2} \tanh(z_{ij})$
   // Round each element to the nearest integer
5: $\tilde{\mathbf{z}}_\mathbf{i} = W_{up-proj} \cdot \text{round}(\mathbf{z_i})$
6: **return** $\tilde{\mathbf{z}}_\mathbf{i}$
---

The product $W_{proj} \, \mathbf{z_i}$ facilitates scalar quantization within a lower-dimensional latent space, thereby defining a compact codebook. This is similar to the low dimensional space used for code index lookup in Yu et al. (2022a). The subsequent up-projection operation then restores the quantized code to its original dimensionality.

**Resampling**  The cross-attention based resampling layer can be used for both down (resp. up) sampling, effectively reduces (resp. increases) the length of the sequence of embeddings is described in Algorithm 2.

---
**Algorithm 2** Resampling Layer with Positional Encoding
---
1: **Input:** `Inputs` $\in \mathbb{R}^{T \times d}$ , `Mask`, `target size: p`, `features dim: d` , `Queries: [Optional]`
2: **Output:** `Queries, Inputs`
3: **If** `Queries = None` **then:**
4:   `Queries` ← `SinPositionalEncoding(p)` $\in \mathbb{R}^{p \times d}$
5: `Queries, Keys, Values` ← `Linear(Queries), Linear(Inputs), Linear(Inputs)`
6: `AttentionWeights` = `Softmax` $\left( \frac{\texttt{Queries} \cdot \texttt{Keys}^t}{\sqrt{d}} * \texttt{Mask} \right) \in \mathbb{R}^{p \times T}$
7: `Output` = `AttentionWeights` $\cdot$ `Values` $\in \mathbb{R}^{p \times d}$
8: `Queries` ← `MLP(Output)`
9: `Inputs` ← `MLP(Inputs)`
10: **return** `(Queries, Inputs)`
---

**Local Cross-Attention Masking**  The encoder network used in this work preserves the notion of residue order, as defined in the primary structure of a protein (i.e. its ordered sequence of amino acids). We do not provide our algorithm with information regarding the amino-acids. However, we do include the order of the residues from which extract the atoms coordinates. When downsampling the encoder representations using a standard cross-attention operation, the resulting output virtually includes information from any other residue embeddings, irrespective of their relative position in the sequence. In order to encourage the downsampled

representation to carry local information, we propose to use local masks in the `CrossAttention` update of the resampling layer defined in Algorithm 2. This will guide the network towards local positions (in the sequence) and prevent information from distant embeddings. We illustrated the local masking in Figure 5, where only the direct neighbors are kept in the attention update.

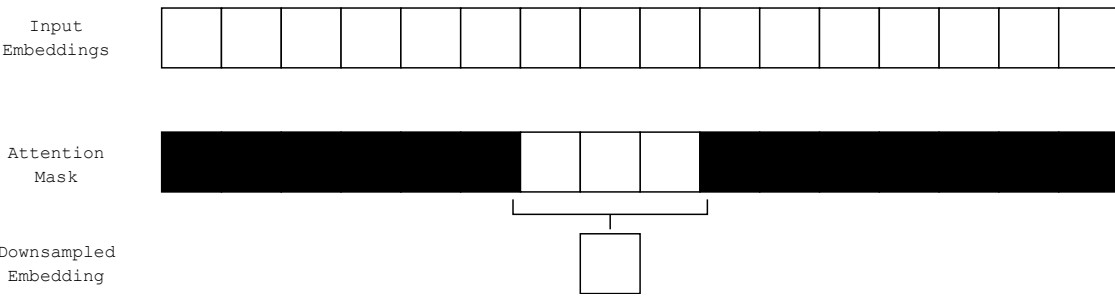

Figure 5: Illustration of the local attention mechanism when using 2 neighbors for aggregation.

**Decoder** For the decoder, we re-purpose the Structure Module (SM) of AlphaFold (Jumper et al., 2021). In Jumper et al. (2021), a pair of 1D and 2D features (called the *single representation* and *pair representation* respectively) is extracted from the data by the Evoformer and fed to the SM to reconstruct the 3D structure. Contrary to AlphaFold, we encode the structures with a set of 1D features - the sequence of discrete codes obtained after tokenization. Inspired by the `OuterProductMean` module of the Evoformer (Alg. 10 in SM of Jumper et al. (2021)), we compute a pairwise representation of the structure by computing the outer product of the quantized sequence after projection, and concatenating the mean with the pair relative positional encoding, as defined in Algorithm 3.

---

**Algorithm 3** Pairwise Module

1: **Input:** $\mathbf{s} = (s_i)_{i \leq N}$
2: **Output:** $\mathbf{k} = (k_{ij})_{i,j \leq N}$
    // linear transforms of the initial embedding
3: $\mathbf{s}^{left} = W_{left}.\mathbf{s}, \qquad \mathbf{s}^{right} = W_{right}.\mathbf{s}$
    // $(n = k)$: protein length, $d$: embedding dim
4: $\mathbf{k} = \texttt{einsum(nd, kd -> nkd}, \mathbf{s}^{left}, \mathbf{s}^{right})$
5: $\mathbf{k} = \text{MLP}(k_{ij}, \text{RelativePositionalEncoding}(i,j)_{i,j \leq N})$
6: **return** $\mathbf{k} = (k_{ij})_{i,j \leq N}$

---

Overall algorithm, the encoding and decoding processes write as described in Algorithm 4

### A.2 Training Details and Metrics

**Data Preprocessing** We consider the graph $\mathcal{G} = (\mathcal{V}, \mathcal{E})$ consisting of a set of vertices - or nodes - $\mathcal{V}$ (the residues) with features $f_V^1 \ldots f_V^{|V|}$ and a set of edges $\mathcal{E}$ with features $f_E^1 \ldots f_E^{|E|}$. For the node features, we use a sinusoidal encoding of sequence position such that for the $i$-th residue, the positional encoding is $(\phi(i,1) \ldots \phi(i,d))$ where $d$ is the embedding size. For the edge features we follow Ganea et al. (2022). More specifically, for each defined by residue $v_i$, a local coordinate system is formed by (a) the unit vector $t_i$ pointing from the $\alpha$-carbon atom to the nitrogen atom, (b) the unit vector $u_i$ pointing from the $\alpha$-carbon to the carbon atom of the carboxyl ($-CO-$) and (c) the normal of the plane defined by $t_i$ and $u_i$: $n_i = \frac{u_i \times t_i}{\|u_i \times t_i\|}$. Finally, setting: $q_i = n_i \times u_i$, the edge features are then defined as the concatenation of the following:

- relative positional edge features: $p_{j \to i} = (n_i^T u_i^T q_i^T)(x_j - x_i)$,

- relative orientation edge features: $q_{j \to i} = (n_i^T u_i^T q_i^T)n_j, \ k_{j \to i} = (n_i^T u_i^T q_i^T)u_j, \ t_{j \to i} = (n_i^T u_i^T q_i^T)v_j,$

---

**Algorithm 4** Overall Algorithm Pseudo-Code

---

1: **Input: $\mathbf{p} \in \mathbb{R}^{N \times 4 \times 3}$, $(\theta, \phi, \psi)$**
2: **Output: $\tilde{\mathbf{z}}, \tilde{\mathbf{p}}$**
   // Compute embedding at the residue-level
3: $\mathbf{z} = \texttt{GNN}(\mathbf{p})$
   // Downsample $N \rightarrow N/r$
4: $\mathbf{z} = \texttt{Resampling}(\mathbf{z})$
   // Quantize
5: $\tilde{\mathbf{z}} = q_\phi(\mathbf{z})$
   // Upsample $N/r \rightarrow N$
6: $\mathbf{s} = \texttt{Resampling}(\tilde{\mathbf{z}})$
   // Make pairwise representation for decoding
7: $\mathbf{k} = \texttt{PairWiseModule}(\mathbf{s})$
   // Decode
8: $\tilde{\mathbf{p}} = \texttt{StructureModule}(\mathbf{s}, \mathbf{k})$
9: **return $\tilde{\mathbf{z}}, \tilde{\mathbf{p}}$** (Quantized output)

---

- distance-based edge features, defined as radial basis functions: $f_{j \rightarrow i,r} = e^{-\frac{\|x_j - x_i\|^2}{2\sigma_r^2}}, r = 1, 2...R$ where $R = 15$ and $\sigma_r = 1.5$.

**Regularization** We found that introducing a scheduled commitment loss, similar to the original VQ-VAE approach van den Oord et al. (2017), significantly improves late-stage stability in training. However, imposing this penalty too early can harm encoder expressivity. To address this, we delay the onset of the commitment loss until step 20000 and then linearly increase its weight from 0 at step 20000 to a maximum value of $\lambda_{\max} = 0.2$. Concretely, the auxiliary loss takes the form:

$$\mathcal{L}_{\text{aux}}(\text{step}, z_i) = \lambda(\text{step}) \left\| z_i - \texttt{stopgrad}(\text{rounded}(z_i)) \right\|, \tag{5}$$

where the schedule function $\lambda(\text{step})$ is given by

$$\lambda(\text{step}) = \begin{cases} 0, & \text{if step} < T_0, \\ \lambda_{\max} \dfrac{\text{step} - T_0}{T_1 - T_0}, & \text{if } T_0 \leq \text{step} \leq T_1, \\ \lambda_{\max}, & \text{if step} > T_1, \end{cases} \tag{6}$$

$T_0$ is the step at which the ramp-up starts and $T_1$ is the step at which the ramp-up finishes. We empirically set $T_0 = 20000$ and $T_1 = 40000$ which we found work well in practice. This delayed, gradually increasing penalty ensures that the encoder can initially learn expressive representations, then become more stable in later training.

**Compression Factor** We define the compression factor of Table 1 as the ratio between the number of bits necessary to encode the atoms of the position of the backbone and the number of latent codes to store multiplied by $\log(\#Codes)$. With positions stored as 64-bit floats, we can write the compression factor as:

$$\text{Compression Factor} = \frac{N \times 4 \times 3 \times 64}{N \times \log_2(\#Codes)/r} = \frac{768}{\log_2(\#Codes)/r} \tag{7}$$

## A.3   Additional Results: Structures Autoencoding

In Figures 6 and 7 we detail the evolution of the RMSD and TM-scores on CASP-15 dataset when varying the downsampling ratio with a fixed codebook size of 4096. The results indicate that the average error increases alongside the standard deviation of the error distribution.

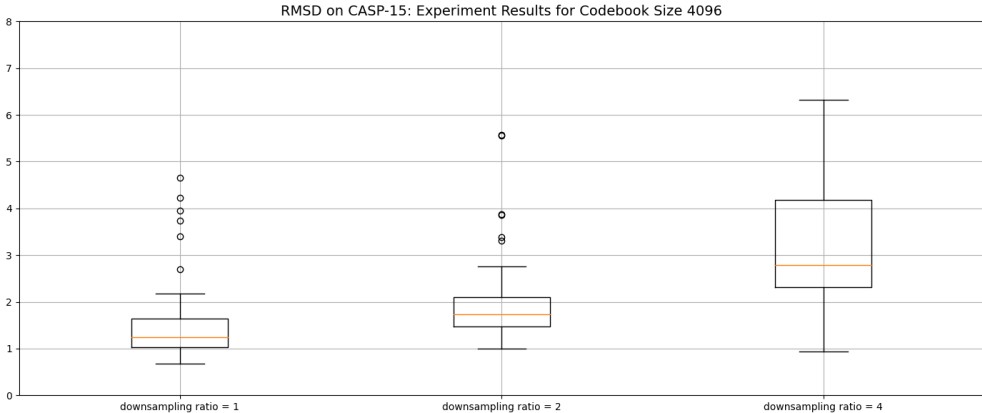

Figure 6: Evolution of the RMSD distribution on CASP-15 dataset with the downsampling ratio, with a fixed codebook size of 4096

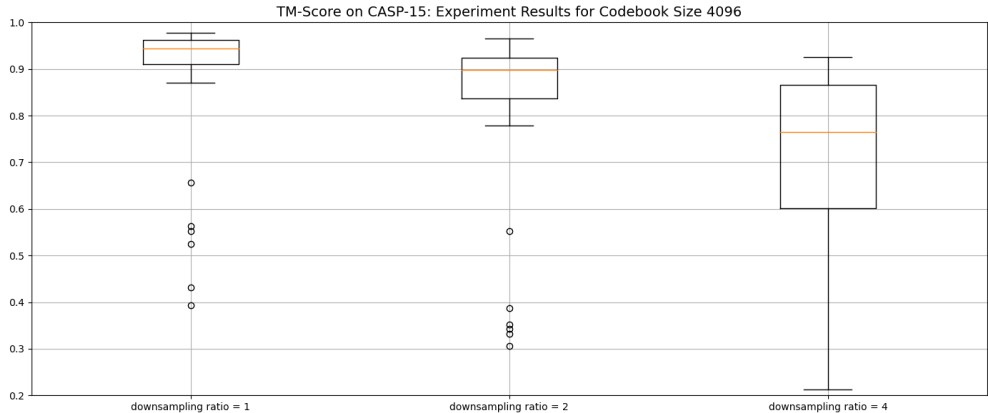

Figure 7: Evolution of the TM-Score distribution on CASP-15 dataset with the downsampling ratio, with a fixed codebook size of 4096

## A.4 De novo Structure Generation

### A.4.1 Training details

**GPT hyperparameters**   We use a standard decoder only transformer following the implementation of (Hoffmann et al., 2022) with pre-layer normalization and a dropout rate of 10% during training. We follow Hoffmann et al. (2022) for the parameters choice with 20 layers, 16 heads per layers, a model dimension of 1024 and a query size of 64, resulting in a model with 344M parameters.

**Training and Optimization**   Given the tokenization of PDB training set, we respectively prepend a `<bos>` and append `<eos>` tokens and pad all sequences with `<pad>` token so that all sequences are of size 514. Hence, the maximum number of actual structural token per sequence is 512. The loss associated to `<pad>` tokens is masked out.

For the optimization, we utilize AdamW with $\beta_1 = 0.95$, $\beta_2 = 0.9$ and a weight decay of 0.1. The learning rate follows a linear warm-up schedule, increasing linearly to $5 \times 10^{-5}$ over the first 1 000 training steps. Following the seminal work of Radford and Narasimhan (2018), we employ embedding, residual, and attention dropout with rate of 10%. We found that batch size has crucial importance in optimizing and adopt a batch size of 65,792 tokens per batch.

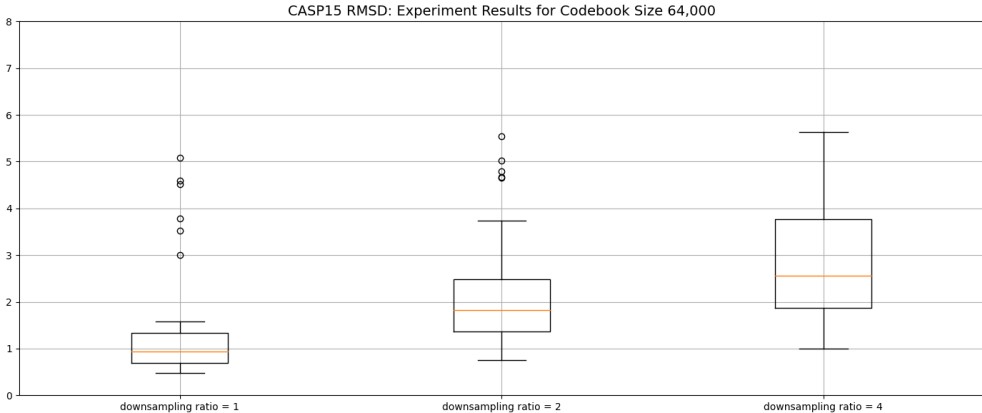

Figure 8: Evolution of the RMSD distribution on CASP-15 dataset with the downsampling ratio, with a fixed codebook size of 4096

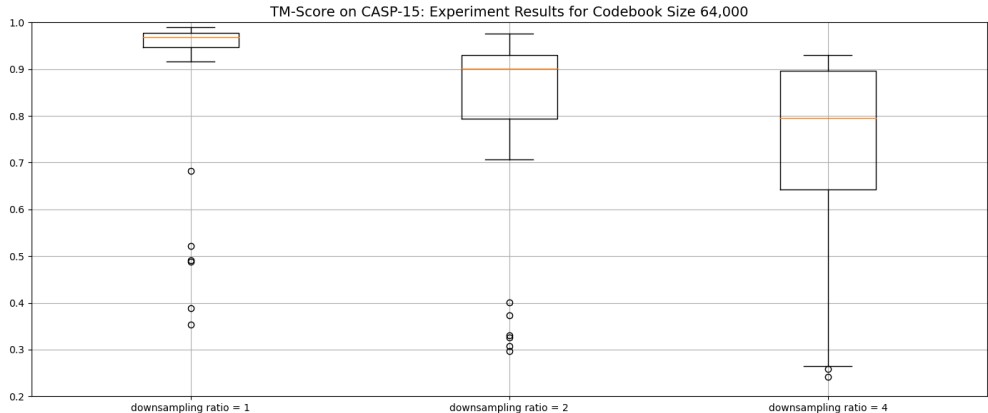

Figure 9: Evolution of the TM-Score distribution on CASP-15 dataset with the downsampling ratio, with a fixed codebook size of 4096

The total number of actual structural tokens is 70M. In that perspective, and in line with work such as Hoffmann et al. (2022), we believe that leveraging a large dataset of predicted structures such as AlphaFold [2] can provide significant improvements when training the latent generative model.

### A.4.2 Structure generation metrics

**Designability**   We adopt the same framework than (Yim et al., 2023; Trippe et al., 2023; Wu et al., 2024) to compute the designability or *self-consistency* score:

- Compute 8 putative sequences from ProteinMPNN (Dauparas et al., 2022) with a temperature sampling of 0.1.

- Fold each of the 8 amino-acid sequences using ESMFold (Lin et al., 2022) without recycling, resulting in 8 folds per generated structure.

- Compare the 8 ESMFold-predicted structures with the original sample using either TM-score (scTM) or RMSD (scRMSD). The final score is taken to be the best score amongst the 8 reconstructed structures.

---

[2] https://alphafold.ebi.ac.uk/

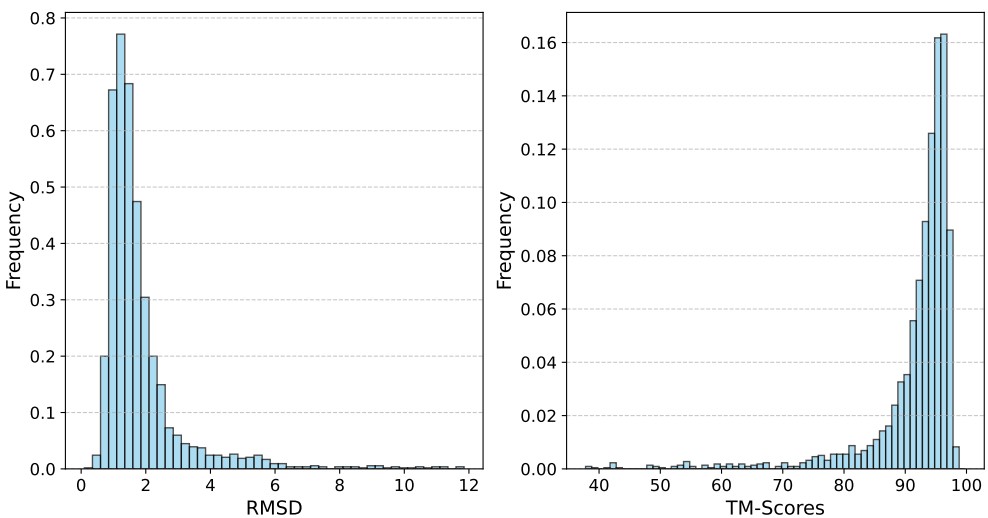

Figure 10: RMSD (left) and TM-Score (right) distribution on the held out test set for the codebook size of 432 and downsampling ratio of 1.

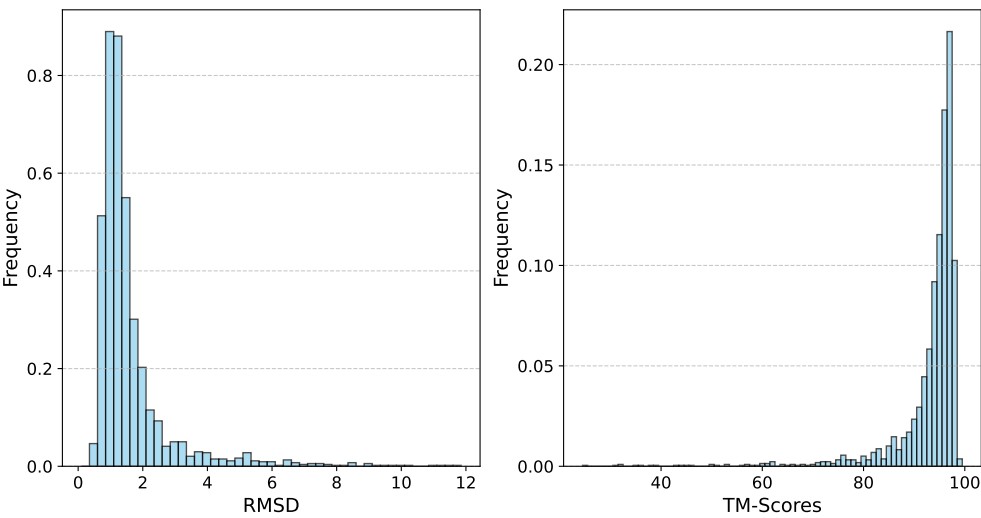

Figure 11: RMSD (left) and TM-Score (right) distribution on the held out test set for the codebook size of 1728 and downsampling ratio of 1.

In Table 2, we report the proportion of generated structures that are said to be designable, *i.e* samples for which scTM>0.5 (or scRMSD< 2 Å when using the RMSD).

**Novelty**    For the reference dataset, we use the s40 CATH dataset (Orengo et al., 1997), publicly available at `ftp://orengoftp.biochem.ucl.ac.uk/cath/releases/latest-release/non-redundant-data-sets/cath-dataset-nonredundant-S40.pdb.tgz`. In order to reduce the computation time, we first retrieve the top $k = 1000$ hits using Foldseek (van Kempen et al., 2024). We then perform TM-align (Zhang and Skolnick, 2005) for each match against the targeted sample and report the TM-score corresponding to the best hit. A

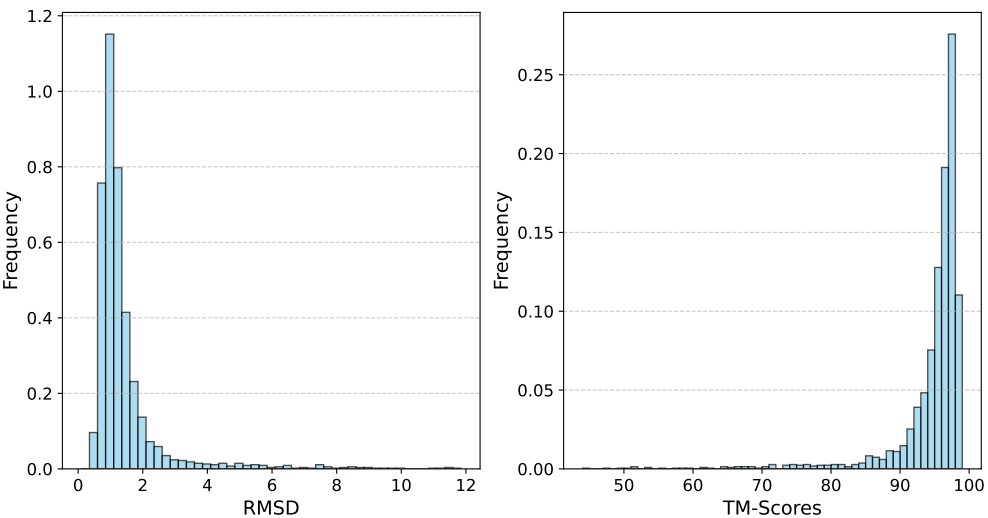

Figure 12: RMSD (left) and TM-Score (right) distribution on the held out test set for the codebook size of 4096 and downsampling ratio of 1.

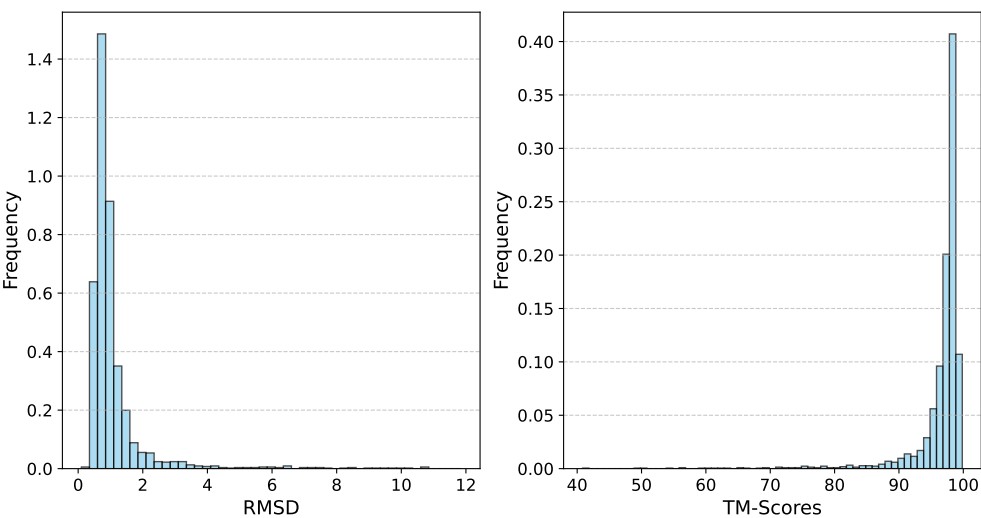

Figure 13: RMSD (left) and TM-Score (right) distribution on the held out test set for the codebook size of 64,000 and downsampling ratio of 1.

structure is then considered as novel if the maximum TM-score against CATH (cathTM) is lower than 0.5 and report the proportion of novel structures in Table 2

**Diversity**  Finally, we measure the diversity of the samples similarly to Watson et al. (2023). More specifically, the generated samples are clustered using an all-to-all pairwise TM-score as the clustering criterion and we observe the resulting number of structural clusters normalized by the number of generated samples. For a diverse set of generated samples, each cluster should be composed of only a few samples - or

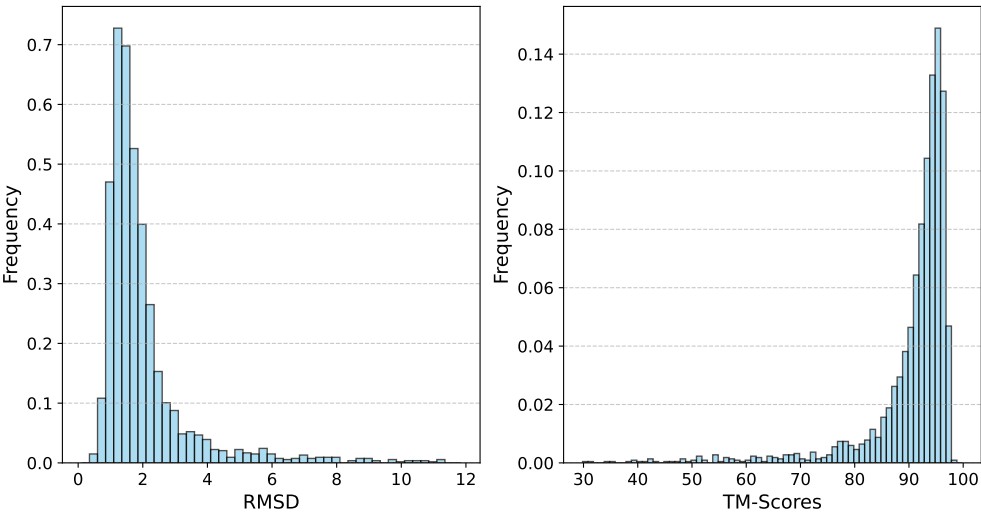

Figure 14: RMSD (left) and TM-Score (right) distribution on the held out test set for the codebook size of 4096 and downsampling ratio of 2.

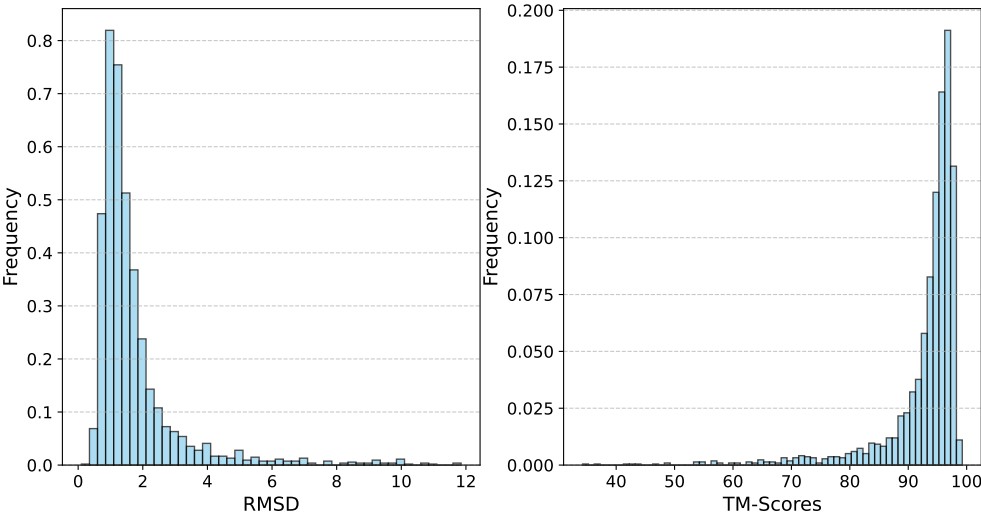

Figure 15: RMSD (left) and TM-Score (right) distribution on the held out test set for the codebook size of 64,000 and downsampling ratio of 2.

equivalently, the number of different clusters should be high. We use MaxCluster Herbert and Sternberg (2008) with a TM-score threshold of 0.6 as in Watson et al. (2023).

### A.4.3 Sampling

**Baselines** For each baseline methods, we follow a standardized process similar to that of Yim et al. (2023) to generate the testing dataset: we sample 8 backbones for every length between 100 and 500 with length step of 5: $[100, 105, \ldots, 500]$. We re-use the publicly available codes and use the parameters reported in Watson et al. (2023) and Yim et al. (2023) respectively.

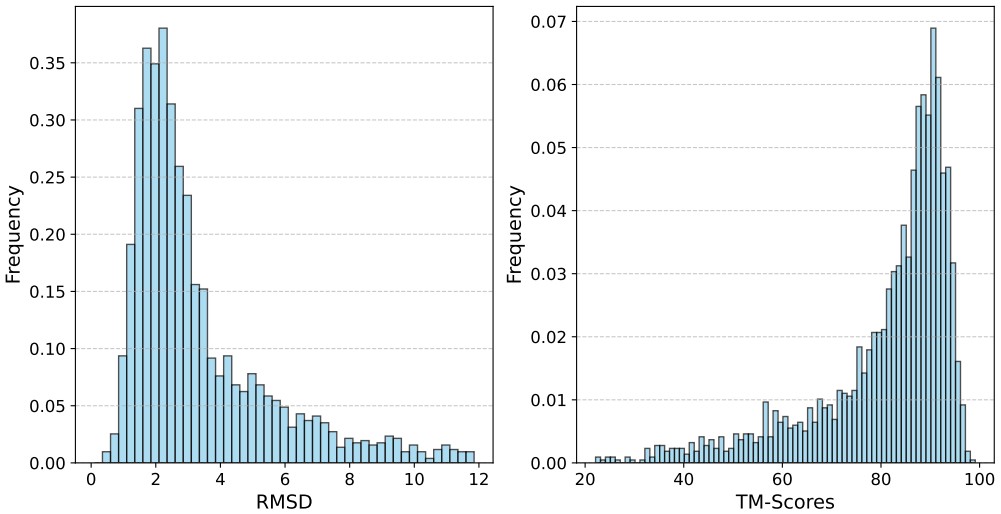

Figure 16: RMSD (left) and TM-Score (right) distribution on the held out test set for the codebook size of 4096 and downsampling ratio of 4.

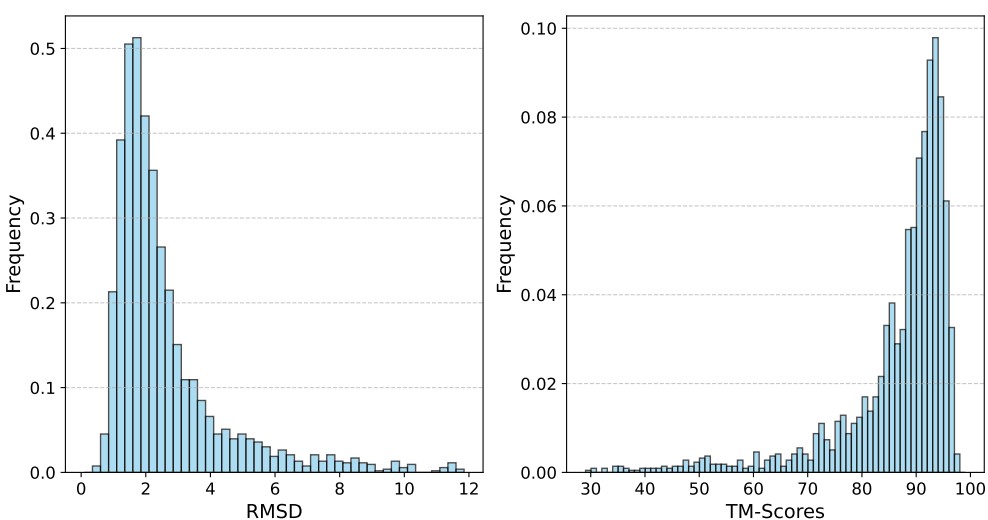

Figure 17: RMSD (left) and TM-Score (right) distribution on the held out test set for the codebook size of 64,000 and downsampling ratio of 4.

**Generating Structures with a Decoder-Only Transformer**   Sampling for our method is a 2 steps process: first, sample a sequence of structural tokens from the trained prior described in Appendix A.4.1, then reconstruct the structures using the trained decoder. There are many ways to sample from a decoder-only transformer model (Vaswani et al., 2017; Radford and Narasimhan, 2018; Radford et al., 2019; Holtzman et al., 2020). We chose temperature sampling (Vaswani et al., 2017) with other alternative sampling strategies such as top-k (Radford et al., 2019) and top-p (or nucleus) sampling (Holtzman et al., 2020) resulting in little improvement at the cost of increased complexity. As showed in Vaswani et al. (2017), the temperature controls the trade-off between the confidence and the diversity of the samples. In order to tune the temperature, we

sampled 2000 samples for each temperature between 0.2 and 0.8 in steps of 0.2 and compute the designability score for these samples. As expected, the higher the temperature, the more varied the samples. Indeed, the distribution of the proteins length, depicted in Figure 18, shows a greater diversity of higher temperatures. On the other hand, with lower temperatures, the length distribution is more concentrated around few lengths. Similarly we can see than for lower temperatures, the samples closer to the modes of the length distribution (samples with length between 100 and approximately 300) achieve higher scTM-score (see Figure 18). The results reported in Table 2 are obtained with a temperature of 0.6, as it achieves a satisfying trade-off between designability and diversity.

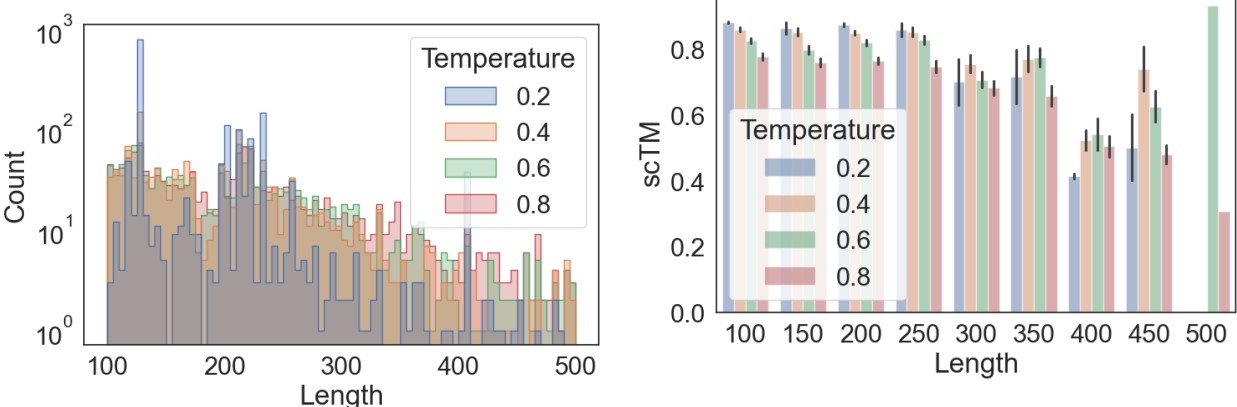

Figure 18: Ablation of the temperature sampling. Left: Histogram of the generated structure length. Right: Designability score vs temperature sampling.

Contrary to the baselines, our model learns the joint distribution the length and the structures $p(s) = \int_l p(s, l) dl$ where the random variables $s$ and $l$ represent the structures and the length respectively. Indeed, only the conditional distribution $p(s|l)$ is modeled by the diffusion-based baselines. In Figure 18, we show the length distribution learned by the model.Since we can only sample from the joint distribution and not the conditional, we adopt the following approach: First, we sample 40,000 structures from the model (using a temperature of 0.6 as previously established). We then bin the generated structures by length, with a bin width of 5 and bin centers uniformly distributed between 100 and 500, specifically: $[100, 105, \dots, 500]$. Finally, we limit the maximum number of structures per bin to 10, randomly selecting 10 structures if a bin contains more than this number.

### A.4.4 Additional Results

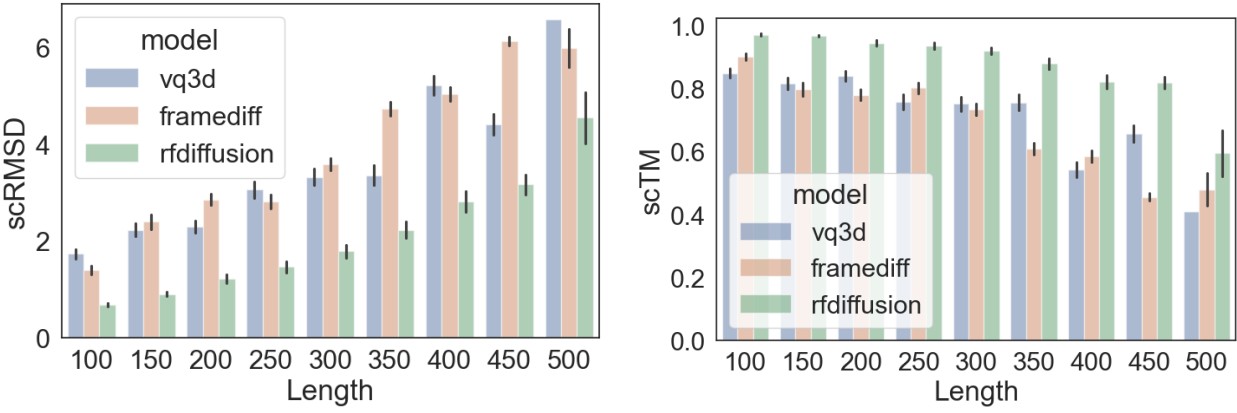

Figure 19: Designability score vs samples length. Left: scRMSD score for different structure lengths. Right: scTM score for different structure lengths.

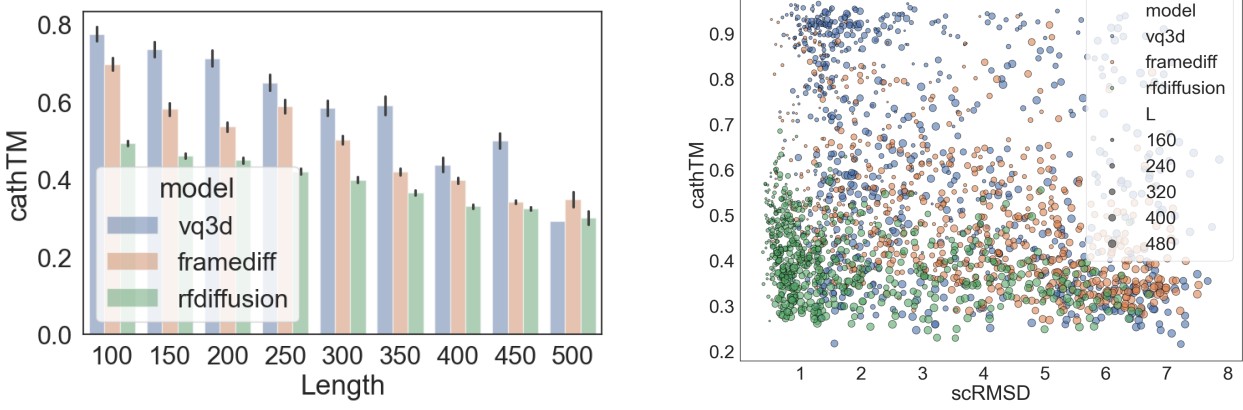

Figure 20: Left: Novelty score for different structure lengths. Right: Novelty score *versus* designability score.

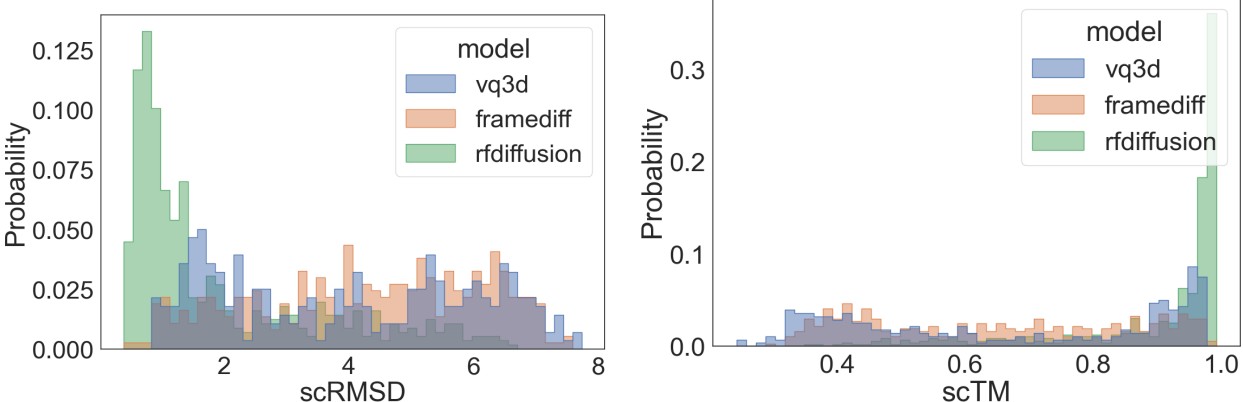

Figure 21: Distribution of the designability scores for novel domains (cathTM<0.5).

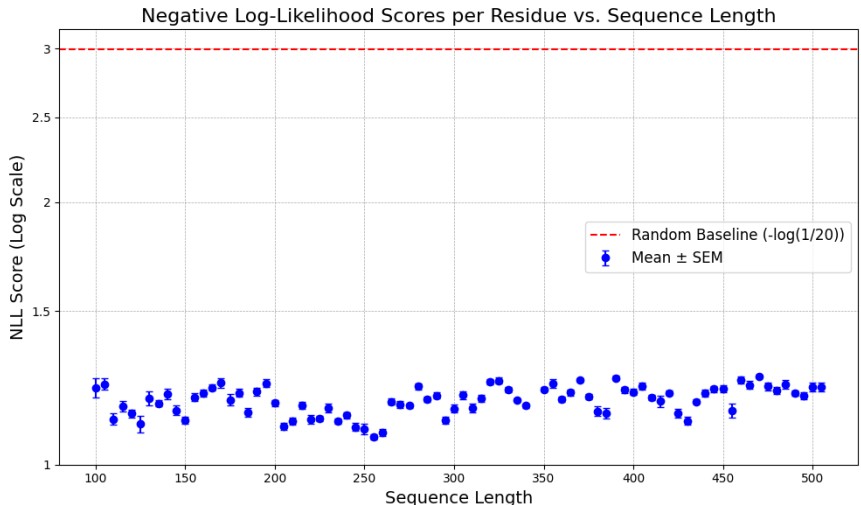

Figure 22: Evolution with the length of the generated sequence of the per-residue negative Log-Likelihood of the selected amino acids as provided by ProteinMPNN

