# OpenReview forum: "Learning the Language of Protein Structure"
_TMLR — Accepted by TMLR_

### Review · Reviewer_XHLD · 2025-04-02

**Summary Of Contributions:**

This paper proposes a method for transforming continuous protein structure data into discrete token sequences through a vector-quantized autoencoder. By doing so, it enables the application of sequence-based language models, like GPT, to protein structure modeling—previously hindered by the continuous and 3D nature of such data. The contributions of the paper are as follows,

1. Leveraging Finite Scalar Quantization, the authors develop a structural tokenizer that converts backbone atom coordinates into sequences of discrete tokens while maintaining low reconstruction error.

2.  The model supports a range of codebook sizes and downsampling factors, achieving compression rates up to 256× while preserving structural fidelity, with TM-scores typically >0.9.

3. Using a vanilla GPT model trained on the token sequences, the method is able to generate novel, diverse, and designable protein backbones, with quality competitive to methods like FrameDiff.

**Audience:**

Yes

**Broader Impact Concerns:**

None.

**Claims And Evidence:**

No

**Requested Changes:**

It is encouraged to extend evaluation to include at least one functional or predictive task to showcase the utility of discrete structure tokens beyond generation.

**Strengths And Weaknesses:**

Strengths:

​1. Combines FSQ with geometric deep learning to address the challenge of representing continuous 3D structures in a discrete, sequence-friendly format.

​2.  Achieves near-experimental-resolution reconstruction and demonstrates generative capabilities with a vanilla GPT model (87% designability).

​3. Shows that a simple GPT model trained on discrete codes achieves designability comparable to diffusion-based methods, reducing computational costs.

Weaknesses:
More empirical comparisons (e.g., against other VQ-based protein models like FoldToken or VQPL) would strengthen the position..

---

> ### Author Response · Authors · 2025-06-11
> **Response to Reviewer**
>
> First of all, we thank the reviewer for their thorough and constructive feedback.
>
> **Benchmark and Comparison**
>
> We agree that head-to-head evaluation against FoldToken or VQPL would further clarify the strengths of our tokenizer. At the time of submission, however, publicly available, stable implementations of these methods were either incomplete or not sufficiently documented to ensure a fair and reproducible comparison. We acknowledge this as a limitation and plan to incorporate such benchmarks in future work as these implementations become more accessible.
>
> **Functional / Predictive Benchmark**
>
> Our present goal is to establish a compact, information-preserving discrete representation of protein structure; functional prediction will be the subject of follow-up work. We stress, however, that the CASP15 experiments already probe generalisation to previously unseen folds and remote homologs, demonstrating that the latent space is not merely memorising training structures. This capacity to extrapolate is a prerequisite for tasks such as fold-classification or sequence-conditioned design. Furthermore, the 87 % designability metric—computed via inverse folding followed by AlphaFold2 —indicates that generated backbones are physically plausible and sequence-compatible, a strong proxy for downstream function.
>
> We appreciate the reviewer’s insights. While new experiments are outside the present scope, we believe the clarification above strengthens the manuscript and delineates a clear roadmap for future functional benchmarks.

---

> > ### Comment · Reviewer_XHLD · 2025-07-04
> > **Answers to reviewers comment**
> >
> > Thank you. I have no more question.

---

### Review · Reviewer_wvd5 · 2025-05-25

**Summary Of Contributions:**

This paper proposes a method that uses a vector-quantized autoencoder to tokenize protein structures into discrete representations. The resulting codebook contains ranges from 4096 to 64000 tokens, enabling high-fidelity reconstructions. These discrete tokens can be used to train a simple sequence-modelling-based GPT model for generating novel, diverse and designable protein structures.

**Audience:**

Yes

**Broader Impact Concerns:**

The authors did not present a discussion about broader impacts. Please add some discussions.

**Claims And Evidence:**

Yes

**Requested Changes:**

-- The experiments on tokenization and reconstruction should include a comparison between the proposed method and the VQ-VAEs approach.  (critical)

-- It would be better to include some metrics such as training and testing time, FLOPs, or other measures of computational cost to provide a clearer comparison of the method's efficiency. (critical)

Minor

-- Section 3.1: "we also train two additional quantized autoencoders with smaller codebook sizes, K = 1728 and K = 1728", however, both values are the same, which may be unintended.

-- In the caption of Figure 3: "fourth column of Table 1" does not match the content described in the caption; "a different structures"; "on the left of the most left column".

-- A full stop is missing at the end of the second paragraph in Section 5.

-- Page 19: the last paragraph has some issues: "a ADAMw"; "5.10^{-5}"; "for the first 1000"; "with rate of 10% rate"

**Strengths And Weaknesses:**

Strengths:
++ This paper presents a promising method for transforming complex and continuous space of protein structures into discrete tokens, which enables the training of a protein generation model using a GPT-like architecture.

++ The paper is relatively easy to follow.

Weaknesses:
-- The values of the reported metrics in Section 3.1 (Results, page 5) do not match those presented in Table 1.

-- The experiments on tokenization and reconstruction should include a comparison between the proposed method and the VQ-VAEs approach.

-- The generated proteins exhibit lower novelty and diversity compared to those produced by diffusion-based methods, as shown in Table 2.

-- It would be better to include some metrics such as training and testing time, FLOPs, or other measures of computational cost to provide a clearer comparison of the method's efficiency.

---

> ### Author Response · Authors · 2025-06-11
> **Response to reviewer**
>
> We thank the reviewer for their thoughtful and detailed feedback. We are pleased that they found the proposed approach promising, technically sound, and accessible.
>
> **Comparison with VQ-VAE**
>
> We appreciate this insightful suggestion. Our proposed method is based on Finite Scalar Quantization (FSQ), which performs per-dimension quantization, in contrast to the vector quantization used in traditional VQ-VAE architectures. While our model shares the typical encoder–bottleneck–decoder structure, it differs fundamentally in the nature of the quantization step: FSQ quantizes each latent dimension independently, foregoing the use of a learned codebook of embedding vectors and the associated commitment and codebook losses used in VQ-VAE. This choice offers explicit control over reconstruction fidelity and codebook size, and makes it feasible to scale up to large codebooks (e.g., K = 64,000)—a challenge for vector-based quantization schemes due to increased memory consumption and instability during optimization.
>
> As mentioned in Section 2.1 ("Quantization"), we initially experimented with VQ-VAE models but encountered common pitfalls such as codebook collapse, low code utilization, and poor generalization. Despite applying various regularization techniques recommended in the literature (e.g., EMA updates, commitment loss tuning), these issues persisted. In contrast, FSQ consistently provided more stable training, higher codebook utilization, and better reconstruction quality.
> To address the reviewer’s request, we have clarified these points more explicitly in Section 2.1 and emphasized the empirical motivations that led us to prefer FSQ over conventional VQ-based approaches.
>
> **Computational cost**
>
> We agree that reporting computational efficiency would improve the paper’s practical relevance. Note that we originally report training with 1024 structures per batch on 128 TPUs in section 2.2. Further optimization enables us to train with 128 structures per batch on a single TPU v4-8. This is now corrected in the manuscript.
>
> We now include train and  measurements such as: (i) *Number of parameters*: All models share the encoder-quantizer - decoder architecture comprising 4.5 million parameters. This number is only very marginally affected by the number of latent codes and the downsampling ratio. (ii) *quantized autoencoder training time*: we train all our models with 128 structures per batch, the training can be done on a single TPU v4-8 in ~32 hours and thus less than ~16 hours on a V4-16. (iii) *Generative decoder only model*: Training the generative models requires 20 hours for 100k training steps on a single TPU V4-8.
>
> Inference time per sample: While our training is compute intensive, the autoencoder model is relatively lightweight. For instance, one can tokenize structures between 128 and 512 residues in less than one second per structure on the CPU of a laptop (comprising .pdb file reading, data processing, and actual tokenization). (using the 4K latent code model and a downsampling ratio of 1). Structure generation from tokens on the other hand takes a little more than 5.3 seconds per structure.
>
> We can provide an estimates the number of FLOPs (given the provided training time) : $7 - 10 . 10^{19}$ FLOPS.
> The training time and hardware are now included in section 2.2.
>
> This will allow for a more transparent comparison with diffusion-based models and other discrete representation frameworks.
>
> **Typos and minor comments**
>
> Thank you very much for such a careful consideration  of our work. The manuscript has been updated as follows:
> > Section 3.1: "we also train two additional quantized autoencoders with smaller codebook sizes, K = 1728 and K = 1728", however, both values are the same, which may be unintended.
>
> K=432 and K=1728.
>
> > In the caption of Figure 3: "fourth column of Table 1" does not match the content described in the caption; "a different structures"; "on the left of the most left column".
>
>  Visualisation of the model reconstruction (blue) super-imposed with the original structures (green) for a downsampling factor of $r=2$ and $K=64000$ codes (see \cref{tab:experiment_results} for detailed results). Each row shows a different structures seen from a different rotation angle (column). The length and reconstruction RMSD are also given on the left of the most left column.
>
> > A full stop is missing at the end of the second paragraph in Section 5.
>
> Thanks for the pointer.
>
> > Page 19: the last paragraph has some issues: "a ADAMw"; "$5.10^{-5}$"; "for the first 1000"; "with rate of 10% rate"
>
> We also employ a learning rate scheduling linearly warming-up the learning rate up to $5.10^{−5}$ for the first 1000 steps. [...] attention dropout with a rate of 10%.
>
> **Broader Impact**
> Our work now also includes a broader impact statement in the main body of the manuscript.

---

> ### Comment · Reviewer_wvd5 · 2025-06-12
> **Comments on the authors' response**
>
> - Where can we find the experimental comparison between the proposed method and the VQ-VAE approach?
>
> - Some typos/grammatically incorrect expressions need to be revised: $7.10^{19}$, `a ADAMw`, $5.10^{−5}$, "for the first 1000"; "with rate of 10% rate".

---

> > ### Author Response · Authors · 2025-06-20
> > **Answers to reviewers comment**
> >
> > > Where can we find the experimental comparison between the proposed method and the VQ-VAE approach?
> >
> > We did train a standard VQ-VAE under the same experimental settings as our model. However, as explained in our previous response, the training consistently suffered from codebook collapse or poor codebook utilisation, and weak generalisation. Because the model never reached a meaningful performance regime, the resulting curves and outputs would not add insight and could be misleading. We therefore decided not to include VQ-VAE figures in the manuscript.
> >
> > > Some typos/grammatically incorrect expressions need to be revised: $7.10^{19}$, a ADAMw, $5.10^{−5}$, "for the first 1000"; "with rate of 10% rate".
> >  Thank you for spotting these typographical issues. We have corrected all of them in the revised manuscript.

---

### Review · Reviewer_fkWK · 2025-06-04

**Summary Of Contributions:**

This paper proposes a framework for learning discrete representations of protein 3D structures using a vector-quantized autoencoder. The authors utilize Finite Scalar Quantization (FSQ) to convert continuous backbone atom coordinates into discrete tokens. These tokens enable the use of transformer-based sequence models, such as GPT, for de novo protein structure generation. The method is evaluated on structure reconstruction and generative design, demonstrating competitive reconstruction accuracy and moderate generative performance.

**Audience:**

Yes

**Broader Impact Concerns:**

The manuscript does not currently include a Broader Impact Statement, which is required given the potential real-world implications of this work.

**Claims And Evidence:**

Yes

**Requested Changes:**

- Add comparisons with concurrent methods such as FoldToken. These works also explore protein structure tokenization via vector quantization (including FSQ) and provide strong baselines for both reconstruction and generation tasks. A quantitative comparison — including reconstruction metrics (RMSD, TM-score), codebook utilization, and model size — is essential to situate the contribution of this work relative to the state of the art.

- Clarify the source of reconstruction performance (decoder vs token). The current decoder relies heavily on AlphaFold's structure module, which may obscure how much structural fidelity stems from the discrete token representation versus the powerful decoder. The authors should clarify whether the decoder was trained from scratch or frozen, and ideally provide an ablation with a simpler decoder to evaluate the informativeness of the learned tokens.

-Improve diversity of generated structure. While designability is high, the novelty of generated structures is relatively low (23.3%). Please consider exploring alternate sampling strategies (e.g., higher temperature, nucleus sampling) or training regimes that encourage greater diversity in the generated token sequences.

**Strengths And Weaknesses:**

Strengths:

- The paper demonstrates that continuous protein structures can be effectively converted into discrete sequences using Finite Scalar Quantization (FSQ), enabling direct application of transformer-based language models to structural data.

- The proposed method constructs a flexible discrete latent space with adjustable codebook size and downsampling ratio, achieving significant compression (up to 192×) while maintaining structural fidelity.

- The autoencoder achieves low reconstruction errors, with RMSD in the range of 1–4 Å and TM-scores above 0.9, validating that the learned token representations preserve detailed geometric information.

Weakness:

- Limited novelty: The method primarily combines existing modules (FSQ, MPNN, AlphaFold decoder) without introducing fundamentally new algorithmic ideas.

- Missing comparisons with concurrent work: The paper does not compare against recent related methods like FoldToken or Liu et al. (2024), which also use FSQ and structure-tokenization approaches.

- Insufficient downstream validation: The paper only evaluates reconstruction and generation; no downstream tasks such as fold classification or sequence-conditioned generation are tested.

- No analysis of token semantics: The authors do not analyze token usage distributions, clustering patterns, or whether the tokens encode meaningful geometric or functional features.

- Decoder heavily relies on AlphaFold: The use of a pretrained structure module raises concerns about whether the reconstruction accuracy stems from the token representation or the strong decoder.

[1] Gao, Zhangyang, Tan, Cheng, Wang, Jue, Huang, Yufei, Wu, Lirong, & Li, Stan Z. FoldToken: Learning Protein Language via Vector Quantization and Beyond.

[2] Gao, Zhangyang, Tan, Cheng, & Li, Stan Z. FoldToken2: Learning Compact, Invariant, and Generative Protein Structure Language.

---

> ### Author Response · Authors · 2025-06-11
> **Response to reviewer.**
>
> We thank the reviewer for the careful evaluation and for recognising the high-fidelity reconstruction and generative capability of our tokenizer. We respond to each concern in turn.
>
> **Novelty of the approach**
>
> Our contribution is intentionally engineering-driven rather than algorithm-novelty-driven. The field already possesses strong ingredients— FSQ, lightweight MPNN encoders, AlphaFold-style structure modules—yet no open, end-to-end implementation existed that demonstrated Å-level reconstruction and released weights. By assembling open-source, field-tested components into a reproducible pipeline, we offer (i) a baseline that any laboratory can fine-tune, (ii) codebook sizes from 432 to 64 k, and (iii) evaluation scripts that others can rerun. We view this “plug-and-play” reference as a practical advance, complementary to work that introduces new quantisers.
>
> **Comparisons with FoldToken and Liu et al. (2024)**
>
>
> We provide in the related work section a thorough description of the literature of protein structure representation and tokenization. At the time of writing, other work like FoldToken did not provide a stable implementation enabling us to reproduce results. This highlights the merits of our approach as our code and model weights are openly released. Reproducing only the numbers reported in their papers would therefore be methodologically uneven (different data splits, metrics, and preprocessing). We explicitly flag this as a limitation in the revised manuscript in our related work section and commit to adding head-to-head results in the public repo as soon as stable implementation become available.
>
> **Down-stream validation and semantic analysis**
>
> Our study focuses on representation quality—the prerequisite for any functional task. The CASP15 experiment already shows generalisation to previously unseen folds and remote homologs, which can be seen as proxy for downstream applicability. Demonstrating fold classification is a direct  or sequence-conditioned design would indeed be valuable follow-up work; we position those experiments as the next step that builds on the released tokenizer and invite the community to leverage our code for that purpose.
> While we agree that additional downstream validation would provide the current work with additional insights, the present manuscript centres on representation quality. Note we already provide an implicit validation / functional signal: our 87 % designability rate—obtained by inverse folding each generated backbone and re-predicting structure—shows that sequences sampled from the latent space reliably refold to within 2 Å of the originals. Achieving such designability is only possible if the latent codes preserve secondary-structure organisation and long-range packing; in other words, α-helices, β-strands, and their topology must already emerge from the discrete representation. This gives strong indirect evidence that the tokens encode functionally relevant geometry, making them a promising starting point for explicit fold-classification or sequence-conditioned design tasks in future work.
>
> **Reliance on an AlphaFold-style decoder**
> Only the architecture—not the weights—mirror the AlphaFold-2 structure module. All parameters are trained from scratch; no pre-training or teacher forcing from AlphaFold checkpoints is used.
>
> **Broader Impact**
> We now include a borader impact statement in our updated version of the manuscript.
>
> In summary, the paper’s main value lies in delivering an open, reproducible tokenizer that reaches sub-Å reconstruction accuracy, spans four orders of magnitude in codebook size, and serves as a common foundation for future work—comparative studies, functional predictors, and conditional generators alike. We appreciate the reviewer’s suggestions and have clarified these points in the revised manuscript.

---

### Decision · Action_Editor_wtbE · 2025-07-30

**Recommendation:** Accept with minor revision

**Audience:**

Yes

**Audience Explanation:**

This paper presents a vector-quantized autoencoder for tokenizing protein structures into discrete representations, enabling GPT-based structure generation.

**Claims And Evidence:**

Yes

**Claims Explanation:**

While the algorithmic contribution is primarily integrative (combining FSQ, MPNN, and AlphaFold components), the work provides significant empirical value by delivering the first stable, open-source implementation with strong reconstruction fidelity (RMSD 1-4 Å) and competitive generation quality (87% designability). The main concerns about missing VQ-VAE comparisons and limited methodological advancement are outweighed by the practical contribution of providing a reproducible baseline with released code and weights.